# Sharper Rates and Flexible Framework for Nonconvex SGD with Client and Data Sampling

## Abstract

We revisit the classical problem of finding an approximately stationary point of the average of $n$ smooth and possibly nonconvex functions. The optimal complexity of stochastic first-order methods in terms of the number of gradient evaluations of individual functions is $\mathcal{O}\left(n + n^{1/2}\varepsilon^{-1}\right)$, attained by the optimal SGD methods SPIDER (Fang et al., 2018) and PAGE (Li et al., 2021), for example, where $\varepsilon$ is the error tolerance. However, i) the big-$\mathcal{O}$ notation hides crucial dependencies on the smoothness constants associated with the functions, and ii) the rates and theory in these methods assume simplistic sampling mechanisms that do not offer any flexibility. In this work we remedy the situation. First, we generalize the PAGE algorithm so that it can provably work with virtually any (unbiased) sampling mechanism. This is particularly useful in federated learning, as it allows us to construct and better understand the impact of various combinations of client and data sampling strategies. Second, our analysis is sharper as we make explicit use of certain novel inequalities that capture the intricate interplay between the smoothness constants and the sampling procedure. Indeed, our analysis is better even for the simple sampling procedure analyzed in the PAGE paper. However, this already improved bound can be further sharpened by a different sampling scheme which we propose. In summary, we provide the most general and most accurate analysis of optimal SGD in the smooth nonconvex regime. Finally, our theoretical findings are supposed with carefully designed experiments.

## 1 Introduction

In this paper, we consider the minimization of the average of $n$ smooth functions (1) in the nonconvex setting in the regime when the number of functions $n$ is very large. In this regime, calculation of the exact gradient can be infeasible and the classical gradient descent method (GD) (Nesterov, 2018) can not be applied. The structure of the problem is generic, and such problems arise in many applications, including machine learning (Bishop & Nasrabadi, 2006) and computer vision (Goodfellow et al., 2016). Problems of this form are the basis of empirical risk minimization (ERM), which is the prevalent paradigm for training supervised machine learning models.

### 1.1 Finite-sum optimization in the smooth nonconvex regime

We consider the finite-sum optimization problem

$$\min_{x \in \mathbb{R}^d} \left\{ f(x) := \frac{1}{n} \sum_{i=1}^{n} f_i(x) \right\}, \tag{1}$$

where $f_i : \mathbb{R}^d \to \mathbb{R}$ is a smooth (and possibly nonconvex) function for all $i \in [n] := \{1, \ldots, n\}$. We are interested in randomized algorithms that find an $\varepsilon$-stationary point of (1) by returning a random point $\widehat{x}$ such that $\mathrm{E}\left[\|\nabla f(\widehat{x})\|^2\right] \leq \varepsilon$. The main efficiency metric of gradient-based algorithms for finding such a point is the (expected) number of gradient evaluations $\nabla f_i$; we will refer to it as the *complexity* of an algorithm.

## 1.2 RELATED WORK

The area of algorithmic research devoted to designing methods for solving the ERM problem (1) in the smooth nonconvex regime is one of the most highly developed and most competitive in optimization.

**The path to optimality.** Let us provide a lightning-speed overview of recent progress. The complexity of GD for solving (1) is $\mathcal{O}\left(n\varepsilon^{-1}\right)$, but this was subsequently improved by more elaborate stochastic methods, including SAGA, SVRG and SCSG (Defazio et al., 2014; Johnson & Zhang, 2013; Lei et al., 2017; Horváth & Richtárik, 2019), which enjoy the better complexity $\mathcal{O}\left(n + n^{2/3}\varepsilon^{-1}\right)$. Further progress was obtained by methods such as SNVRG and Geom-SARAH (Zhou et al., 2018; Horváth et al., 2020), improving the complexity to $\widetilde{\mathcal{O}}\left(n + n^{1/2}\varepsilon^{-1}\right)$. Finally, the methods SPIDER, SpiderBoost, SARAH and PAGE (Fang et al., 2018; Wang et al., 2019; Nguyen et al., 2017; Li et al., 2021), among others, shaved-off certain logarithmic factors and obtained the *optimal* complexity $\mathcal{O}\left(n + n^{1/2}\varepsilon^{-1}\right)$, matching lower bounds (Li et al., 2021).

**Optimal, but hiding a secret.** While it may look that this is the end of the road, the starting point of our work is the observation that *the big-$\mathcal{O}$ notation in the above results hides important and typically very large data-dependent constants.* For instance, it is rarely noted that the more precise complexity of GD is $\mathcal{O}\left(L_- n\varepsilon^{-1}\right)$, while the complexity of the optimal methods, for instance PAGE, is $\mathcal{O}\left(n + L_+ n^{1/2}\varepsilon^{-1}\right)$, where $L_- \leq L_+$ are *different* and often *very large* smoothness constants. Moreover, it is easy to generate examples of problems (see Example 1) in which the ratio $L_+/L_-$ is *as large one desires.*

**Client and data sampling in federated learning.** Furthermore, several modern applications, notably federated learning (Konečný et al., 2016; McMahan et al., 2017), depend on elaborate *client* and *data sampling* mechanisms, which are not properly understood. However, optimal SGD methods were considered in combination with very simple mechanisms only, such as sampling a random function $f_i$ several times independently with replacement (Li et al., 2021). We thus believe that an in-depth study of sampling mechanisms for optimal methods will be of interest to the federated learning community. There exists prior work on analyzing non-optimal SGD variants with flexible mechanisms For example, using the "arbitrary sampling" paradigm, originally proposed by Richtárik & Takáč (2016) in the study of randomized coordinate descent methods, Horváth & Richtárik (2019) and Qian et al. (2021) analyzed SVRG, SAGA, and SARAH methods, and showed that it is possible to improve the dependence of these methods on the smoothness constants via carefully crafted sampling strategies. Further, Zhao & Zhang (2014) investigated the stratified sampling, but only provided the analysis for vanilla SGD, and in the convex case.

## 1.3 SUMMARY OF CONTRIBUTIONS

• Specifically, in the original paper (Li et al., 2021), the optimal (w.r.t. $n$ and $\varepsilon$) optimization method PAGE was analyzed with a simple uniform mini-batch sampling with replacement. We analyze PAGE with virtually any (unbiased) sampling mechanism using a novel Assumption 4. Moreover, we show that some samplings can improve the convergence rate $\mathcal{O}\left(n + L_+ n^{1/2}\varepsilon^{-1}\right)$ of PAGE (see Table 2).
• We improve the analysis of PAGE using a new quantity, the weighted Hessian Variance $L_\pm$ (or $L_{\pm,w}$), that is well-defined if the functions $f_i$ are $L_i$–smooth. We show that, when the functions $f_i$ are "similar" in the sense of the weighted Hessian Variance, PAGE enjoys faster convergence rates (see Table 2). Also, unlike (Szlendak et al., 2021), we introduce weights $w_i$ that can play a crucial role in some samplings. Moreover, the experiments in Sec 5 agree with our theoretical results.
• Our framework is flexible and can be generalized to *the composition of samplings*. These samplings naturally emerge in federated learning (Konečný et al., 2016; McMahan et al., 2017), and we show that our framework can be helpful in the analysis of problems from federated learning.

## 2 ASSUMPTIONS

We need the following standard assumptions from nonconvex optimization.

**Assumption 1.** *There exists $f^* \in \mathbb{R}$ such that $f(x) \geq f^*$ for all $x \in \mathbb{R}^d$.*

**Assumption 2.** *There exists $L_- \geq 0$ such that $\|\nabla f(x) - \nabla f(y)\| \leq L_- \|x - y\|$ for all $x, y \in \mathbb{R}^d$.*

Table 1: The constants $A$, $B$, $w_i$ and $|\mathbf{S}|$ that characterize the samplings in Assumption 4.

| Sampling scheme | $A$ | $w_i$ | $B$ | $|\mathbf{S}|$ | Reference |
|---|---|---|---|---|---|
| *Uniform With Replacement* | $1/\tau$ | $1/n$ | $1/\tau$ | $\leq \tau$ | Sec. E.3 |
| *Importance* | $1/\tau$ | $q_i$ | $1/\tau$ | $\leq \tau$ | Sec. E.3 |
| *Nice* | $\frac{n-\tau}{\tau(n-1)}$ | $1/n$ | $\frac{n-\tau}{\tau(n-1)}$ | $\tau$ | Sec. E.1 |
| *Independent* | $\frac{1}{\sum_{i=1}^n \frac{p_i}{1-p_i}}$ | $\frac{\frac{p_i}{1-p_i}}{\sum_{i=1}^n \frac{p_i}{1-p_i}}$ | $0$ | $\sum_{i=1}^n p_i$ | Sec. E.2 |
| *Extended Nice* | $\frac{n-\tau}{\tau(n-1)}$ | $\frac{l_i}{\sum_{i=1}^n l_i}$ | $\frac{n-\tau}{\tau(n-1)}$ | $\leq \tau$ | Sec. E.4 |

Notation: $n$ = # of data points; $\tau$ = batch size; $q_i$ = probability to sample $i^{\text{th}}$ data point in the multinomial distribution; $p_i$ = probability to sample $i^{\text{th}}$ data point in the bernoulli distribution; $l_i$ = # of times to repeat $i^{\text{th}}$ data point before apply the *Nice* sampling.

**Assumption 3.** *For all $i \in [n]$, there existsa constant $L_i > 0$ such that $\|\nabla f_i(x) - \nabla f_i(y)\| \leq L_i \|x - y\|$ for all $x, y \in \mathbb{R}^d$.*

### 2.1 TIGHT VARIANCE CONTROL OF GENERAL SAMPLING ESTIMATORS

In Algorithm 1 (a generalization of PAGE), we form an estimator of the gradient $\nabla f$ via subsampling. In our search for achieving the combined goal of providing a *general* (in terms of the range of sampling techniques we cater for) and *refined* (in terms of the sharpness of our results, even when compared to known results using the *same* sampling technique) analysis of PAGE, we have identified several powerful tools, the first of which is Assumption 4.

Let $\mathcal{S}^n := \{(w_1, \ldots, w_n) \in \mathbb{R}^n \,|\, w_1, \ldots, w_n \geq 0, \sum_{i=1}^n w_i = 1\}$ be the *standard simplex* and $(\Omega, \mathcal{F}, \mathbf{P})$ a probability space.

**Assumption 4** (Weighted $AB$ Inequality). *Consider the random mapping $\mathbf{S} : \mathbb{R}^d \times \cdots \times \mathbb{R}^d \times \Omega \to \mathbb{R}^d$, which we will call "sampling", such that $\mathrm{E}\left[\mathbf{S}(a_1, \ldots, a_n; \omega)\right] = \frac{1}{n} \sum_{i=1}^n a_i$ for all $a_1, \ldots, a_n \in \mathbb{R}^d$. Assume that there exist $A, B \geq 0$ and weights $(w_1, \ldots, w_n) \in \mathcal{S}^n$ such that*

$$\mathrm{E}\left[\left\|\mathbf{S}(a_1, \ldots, a_n; \omega) - \frac{1}{n}\sum_{i=1}^n a_i\right\|^2\right] \leq \frac{A}{n}\sum_{i=1}^n \frac{1}{nw_i}\|a_i\|^2 - B\left\|\frac{1}{n}\sum_{i=1}^n a_i\right\|^2, \quad \forall a_1, \ldots, a_n \in \mathbb{R}^d. \quad (2)$$

For simplicity, we denote $\mathbf{S}\left(\{a_i\}_{i=1}^n\right) := \mathbf{S}(a_1, \ldots, a_n) := \mathbf{S}(a_1, \ldots, a_n; \omega)$. Further, the collection of samplings satisfying Assumption 4 will be denotes as $\mathbb{S}(A, B, \{w_i\}_{i=1}^n)$. The main purpose of a sampling $\mathbf{S} \in \mathbb{S}(A, B, \{w_i\}_{i=1}^n)$ is to estimate the mean $\frac{1}{n}\sum_{i=1}^n a_i$ using some random subsets (possibly containing some elements more than once) of the set $\{a_1, \ldots, a_n\}$. Assumption 4 is the only nonstandard assumption in our paper, and we refer to Table 1, where we provide examples of samplings that satisfy this assumption. It represents a convenient framework to build the theory.

We now define the cardinality $|\mathbf{S}|$ of a sampling $\mathbf{S} \in \mathbb{S}(A, B, \{w_i\}_{i=1}^n)$.

**Definition 1** (Cardinality of a Sampling). Let us take $\mathbf{S} \in \mathbb{S}(A, B, \{w_i\}_{i=1}^n)$, and define the function $\mathbf{S}_\omega(a_1, \ldots, a_n) : \mathbb{R}^d \times \cdots \times \mathbb{R}^d \to \mathbb{R}^d$ such that $\mathbf{S}_\omega(a_1, \ldots, a_n) := \mathbf{S}(a_1, \ldots, a_n; \omega)$. If the function $\mathbf{S}_\omega(a_1, \ldots, a_n)$ depends only on a subset $\mathcal{A}(\omega)$ of the arguments $(a_1, \ldots, a_n)$, where $\mathcal{A}(\omega) : \Omega \to 2^{\{a_1, \ldots, a_n\}}$, we define $|\mathbf{S}| := \mathrm{E}\left[|\mathcal{A}(\omega)|\right]$.

Assumption 4 is most closely related to two independent works: (Horváth & Richtárik, 2019) and (Szlendak et al., 2021). Horváth & Richtárik (2019) analyzed several non-optimal SGD methods for "arbitrary samplings"; these are random set-valued mappings $S$ with values being the subsets of $[n]$. The distribution of a such a sampling is uniquely determined by assigning probabilities to all $2^n$ subsets of $[n]$. In particular, they show that Assumption 4 holds with $\mathbf{S}(a_1, \ldots, a_n) = \frac{1}{n}\sum_{i\in S} \frac{a_i}{p_i}$, $p_i := \mathbf{Prob}(i \in S)$, $|\mathbf{S}| = |S|$, some $A \geq 0$, $w_1, \ldots, w_n \geq 0$ and $B = 0$. Recently, Szlendak et al. (2021) studied a similar inequality, but in the context of communication-efficient distributed training with randomized gradient compression operators. They explicitly set out to study *correlated* compressors, and for this reason introduced the second term in the right hand side; i.e., they considered

Table 2: The complexity of methods and samplings from Table 1 and Sec 4.

| Sampling scheme | Complexity | Comment |
|---|---|---|
| *Independent* (Horváth & Richtárik, 2019) | $\Theta\left(n + \frac{n^{2/3}\left(\frac{1}{n}\sum_{i=1}^{n}L_i\right)}{\varepsilon}\right)$ | SVRG method $p_i \propto L_i$ |
| *Uniform With Replacement* (Li et al., 2021) | $\Theta\left(n + \frac{\sqrt{n}L_+}{\varepsilon}\right)$ | — |
| *Uniform With Replacement* (new) | $\Theta\left(n + \frac{\max\{\sqrt{n}L_\pm, L_-\}}{\varepsilon}\right)$ | — |
| *Importance* | $\Theta\left(n + \frac{\sqrt{n}\left(\frac{1}{n}\sum_{i=1}^{n}L_i\right)}{\varepsilon}\right)$ | $q_i = \frac{L_i}{\sum_{i=1}^{n}L_i}$ |
| *Stratified* | $\Theta\left(n + \frac{\max\left\{\sqrt{n}\sqrt{\frac{1}{g}\sum_{i=1}^{g}L_{i,\pm}^2}, gL_-\right\}}{\varepsilon}\right)$ | The functions $f_i$ are splitted into $g$ groups |

Notation: $n$ = # of data points; $\varepsilon$ = error tolerance; $L_-, L_i, L_\pm, L_+$ and $L_{i,\pm}$ are smoothness constants such that $L_- \leq \frac{1}{n}\sum_{i=1}^{n}L_i$, $L_- \leq L_+$ and $L_\pm \leq L_+$; $g$ = # of groups in the *Stratified* sampling.

the possibility of $B$ being nonzero, as in this way they obtain a tighter inequality, which they can use in their analysis. However, their inequality only involves uniform weights $\{w_i\}$. Our Assumption 4 offers the tightest known way to control of the variance of the sampling estimator, and our analysis can take advantage of it. See Table 1 for an overview of several samplings and the values $A, B$ and $\{w_i\}$ for which Assumption 4 is satisfied.

## 2.2 SAMPLING-DEPENDENT SMOOTHNESS CONSTANTS

We now define two smoothness constants that depend on the weights $\{w_i\}_{i=1}^{n}$ of a sampling $\mathbf{S}$ and on the functions $f_i$.

**Definition 2.** Given a sampling $\mathbf{S} \in \mathbb{S}(A, B, \{w_i\}_{i=1}^{n})$, let $L_{+,w}$ be a constant for which

$$\frac{1}{n}\sum_{i=1}^{n}\frac{1}{nw_i}\|\nabla f_i(x) - \nabla f_i(y)\|^2 \leq L_{+,w}^2\|x - y\|^2, \quad \forall x, y \in \mathbb{R}^d.$$

For $(w_1, \ldots, w_n) = (1/n, \ldots, 1/n)$, we define $L_+ := L_{+,w}$.

**Definition 3.** Given a sampling $\mathbf{S} \in \mathbb{S}(A, B, \{w_i\}_{i=1}^{n})$, let $L_{\pm,w}$ be a constant for which

$$\frac{1}{n}\sum_{i=1}^{n}\frac{1}{nw_i}\|\nabla f_i(x) - \nabla f_i(y)\|^2 - \|\nabla f(x) - \nabla f(y)\|^2 \leq L_{\pm,w}^2\|x - y\|^2, \qquad \forall x, y \in \mathbb{R}^d.$$

For $(w_1, \ldots, w_n) = (1/n, \ldots, 1/n)$, we define $L_\pm := L_{\pm,w}$.

One can interpret Definition 2 as *weighted* mean-squared smoothness property (Arjevani et al., 2019), and Definition 3 as *weighted* Hessian variance (Szlendak et al., 2021) that captures the similarity between the functions $f_i$. The constants $L_{+,w}$ and $L_{\pm,w}$ help us better to understand the structure of the optimization problem (1) *in connection* with a particular choice of a sampling scheme. Note that Definitions 2, 3 and Assumption 4 are connected with the weights $\{w_i\}_{i=1}^{n}$.

The next result states that $L_{+,w}^2$ and $L_{\pm,w}^2$ are finite provided the functions $f_i$ are $L_i$–smooth for all $i \in [n]$.

**Theorem 4.** *If Assumption 3 holds, then* $L_{+,w}^2 = L_{\pm,w}^2 = \frac{1}{n}\sum_{i=1}^{n}\frac{1}{nw_i}L_i^2$ *satisfy Def. 2 and 3.*

Indeed, from Assumption 3 and the inequality $\|\nabla f(x) - \nabla f(y)\|^2 \geq 0$ we get

$$\frac{1}{n}\sum_{i=1}^{n}\frac{1}{nw_i}\|\nabla f_i(x) - \nabla f_i(y)\|^2 - \|\nabla f(x) - \nabla f(y)\|^2 \leq \left(\frac{1}{n}\sum_{i=1}^{n}\frac{1}{nw_i}L_i^2\right)\|x - y\|^2,$$

thus we can take $L_{\pm,w}^2 = \frac{1}{n}\sum_{i=1}^{n}\frac{1}{nw_i}L_i^2$. The proof for $L_{+,w}^2$ is the same.

From the proof, one can see that we ignore $\|\nabla f(x) - \nabla f(y)\|^2$ when estimating $L_{\pm,w}^2$. However, by doing that, the obtained result is not tight.

---

**Algorithm 1** PAGE

---

1: **Input:** initial point $x^0 \in \mathbb{R}^d$, stepsize $\gamma > 0$, probability $p \in (0, 1]$
2: $g^0 = \nabla f(x^0)$
3: **for** $t = 0, 1, \ldots, T$ **do**
4:     $x^{t+1} = x^t - \gamma g^t$
5:     Generate a random sampling function $\mathbf{S}^t$
6:     $g^{t+1} = \begin{cases} \nabla f(x^{t+1}) & \text{with probability } p \\ g^t + \mathbf{S}^t \left( \{\nabla f_i(x^{t+1}) - \nabla f_i(x^t)\}_{i=1}^n \right) & \text{with probability } 1-p \end{cases}$
7: **end for**

---

## 3   A GENERAL AND REFINED THEORETICAL ANALYSIS OF PAGE

In the Algorithm 1, we provide the description of the PAGE method. The choice of PAGE as the base method is driven by the simplicity of the proof in the original paper. However, we believe that other methods, including SPIDER and SARAH, can also admit samplings from Assumption 4.

In this section, we provide theoretical results for Algorithm 1. Let us define $\Delta_0 := f(x^0) - f^*$.

**Theorem 5.** *Suppose that Assumptions 1, 2, 3 hold and the samplings $\mathbf{S}^t \in \mathbb{S}(A, B, \{w_i\}_{i=1}^n)$. Then Algorithm 1 (PAGE) has the convergence rate $\mathrm{E}\left[\left\|\nabla f(\widehat{x}^T)\right\|^2\right] \leq \frac{2\Delta_0}{\gamma T}$, where $\gamma \leq \left(L_- + \sqrt{\frac{1-p}{p}\left((A-B)L_{+,w}^2 + BL_{\pm,w}^2\right)}\right)^{-1}$.*

To reach an $\varepsilon$-stationary point, it is enough to do

$$T := \frac{2\Delta_0}{\varepsilon}\left(L_- + \sqrt{\frac{1-p}{p}\left((A-B)L_{+,w}^2 + BL_{\pm,w}^2\right)}\right) \tag{3}$$

iterations of Algorithm 1. To deduce the gradient complexity, we provide the following corollary.

**Corollary 1.** *Suppose that the assumptions of Thm 5 hold. Let us take $p = \frac{|\mathbf{S}|}{|\mathbf{S}|+n}$. Then the complexity (the expected number of gradient calculations $\nabla f_i$) of Algorithm 1 equals*

$$N := \Theta\left(n + |\mathbf{S}|T\right) = \Theta\left(n + \frac{\Delta_0}{\varepsilon}|\mathbf{S}|\left(L_- + \sqrt{\frac{n}{|\mathbf{S}|}\left((A-B)L_{+,w}^2 + BL_{\pm,w}^2\right)}\right)\right).$$

*Proof.* At each iteration, the expected # gradient calculations equals $pn + (1-p)|\mathbf{S}| \leq 2|\mathbf{S}|$. Thus the total expected number of gradient calculations equals $n + 2|\mathbf{S}|T$ to get an $\varepsilon$-stationary point. $\square$

The original result from (Li et al., 2021) states that the complexity of PAGE with batch size $\tau$ is

$$N_{\mathrm{orig}} := \Theta\left(n + \frac{\Delta_0}{\varepsilon}\tau\left(L_- + \frac{\sqrt{n}}{\tau}L_+\right)\right) \geq \Theta\left(n + \frac{\Delta_0\sqrt{n}L_+}{\varepsilon}\right) \tag{4}$$

for all $\tau \in \{1, 2, \ldots, n\}$.

### 3.1   *Uniform With Replacement* SAMPLING

Let us do a sanity check and substitute the parameters of the sampling that the original paper uses. We take the *Uniform With Replacement* sampling (see Sec E.3) with batch size $\tau$ (note that $\tau \geq |\mathbf{S}|$), $A = B = 1/\tau$ and $w_i = 1/n$ for all $i \in [n]$ (see Table 1) and get the complexity $N_{\mathrm{uniform}} = \Theta\left(n + \frac{\Delta_0}{\varepsilon}\tau\left(L_- + \frac{\sqrt{n}}{\tau}L_\pm\right)\right)$ for all $\tau \in \{1, 2, \ldots, n\}$. Next, let us fix $\tau \leq \max\left\{\frac{\sqrt{n}L_\pm}{L_-}, 1\right\}$, and, finally, obtain that $N_{\mathrm{uniform}} = \Theta\left(n + \frac{\Delta_0 \max\{\sqrt{n}L_\pm, L_-\}}{\varepsilon}\right)$. Let us compare it with (4). With the **same sampling**, our analysis provides better complexity; indeed, note that $\max\{\sqrt{n}L_\pm, L_-\} \leq \sqrt{n}L_+$ (see Lemma 2 in Szlendak et al. (2021)). Moreover, Szlendak et al. (2021) provides examples of the optimization problems when $L_\pm$ is small and $L_+$ is large, so the difference can be arbitrary large.

### 3.2 *Nice* SAMPLING

Next, we consider the *Nice* sampling (see Sec E.1) and get that the complexity $N_{\text{nice}} = \Theta\left(n + \frac{\Delta_0}{\varepsilon}\tau\left(L_- + \frac{1}{\tau}\sqrt{\frac{n(n-\tau)}{(n-1)}}L_\pm\right)\right)$. Unlike the *Uniform With Replacement* sampling, for $\varepsilon$ small enough, the *Nice* sampling recovers the complexity of GD for $\tau = n$, which is equal to $\Theta\left(\frac{\Delta_0 n L_-}{\varepsilon}\right)$.

### 3.3 *Importance* SAMPLING

Let us consider the *Importance* sampling (see Sec E.3) that justifies the introduction of the weights $w_i$. We can get the complexity $N_{\text{importance}} = \Theta\left(n + \frac{\Delta_0}{\varepsilon}\tau\left(L_- + \frac{\sqrt{n}}{\tau}L_{\pm,w}\right)\right) \leq \Theta\left(n + \frac{\Delta_0 \max\{\sqrt{n}L_{\pm,w}, L_-\}}{\varepsilon}\right)$ for $\tau \leq \max\left\{\frac{\sqrt{n}L_{\pm,w}}{L_-}, 1\right\}$. Now, we take $q_i = w_i = \frac{L_i}{\sum_{i=1}^n L_i}$ and use the results from Sec F to obtain $N_{\text{importance}} = \Theta\left(n + \frac{\Delta_0\sqrt{n}\left(\frac{1}{n}\sum_{i=1}^n L_i\right)}{\varepsilon}\right) \leq N_{\text{orig}}$ (See Sec G). In Example 2, we consider the optimization task where $\frac{1}{n}\sum_{i=1}^n L_i$ is $\sqrt{n}$ times smaller than $L_+$. Thus the complexity $N_{\text{importance}}$ can be at least $\sqrt{n}$ times smaller that the complexity $N_{\text{orig}}$.

### 3.4 THE POWER OF $B > 0$

In all previous examples, the constant $A = B > 0$. If $A = B$, then the complexity $N = \Theta\left(n + \frac{\Delta_0}{\varepsilon}|\mathbf{S}|\left(L_- + \sqrt{\frac{n}{|\mathbf{S}|}BL_{\pm,w}^2}\right)\right)$, thus the complexity $N$ does not depend on $L_{+,w}^2$, which greater of equal to $L_{\pm,w}^2$. This is the first analysis of optimal SGD, which uses $B > 0$.

### 3.5 ANALYSIS UNDER PŁ CONDITION

The previous results can be extended to the optimization problems that satisfy the Polyak-Łojasiewicz condition. Under this assumption, Algorithm 1 enjoys a linear convergence rate.

**Assumption 5.** *There exists $\mu > 0$ such that the function $f$ satisfy (Polyak-Łojasiewicz) PŁ-condition:*

$$\|\nabla f(x)\|^2 \geq 2\mu(f(x) - f^*) \quad \forall x \in \mathbb{R},$$

*where $f^* = \inf_{x \in \mathbb{R}^d} f(x) > -\infty$.*

Using Assumption 5, we can improve the convergence rate of PAGE.

**Theorem 6.** *Suppose that Assumptions 1, 2, 3, 5 and the samplings $\mathbf{S}^t \in \mathbb{S}(A, B, \{w_i\}_{i=1}^n)$. Then Algorithm 1 (PAGE) has the convergence rate $\mathrm{E}\left[f(x^T)\right] - f^* \leq (1 - \gamma\mu)^T \Delta_0$, where*

$$\gamma \leq \min\left\{\left(L_- + \sqrt{\frac{2(1-p)}{p}\left((A-B)L_{+,w}^2 + BL_{\pm,w}^2\right)}\right)^{-1}, \frac{p}{2\mu}\right\}.$$

## 4 COMPOSITION OF SAMPLINGS: APPLICATION TO FEDERATED LEARNING

In Sec 3, we analyze the PAGE method with samplings that satisfy Assumption 4. Now, let us assume that the functions $f_i$ have the finite-sum form, i.e., $f_i(x) := \frac{1}{m_i}\sum_{j=1}^{m_i} f_{ij}(x)$, thus we an optimization problem

$$\min_{x \in \mathbb{R}^d}\left\{f(x) := \frac{1}{n}\sum_{i=1}^n \frac{1}{m_i}\sum_{j=1}^{m_i} f_{ij}(x)\right\}, \tag{5}$$

Another way to get the problem is to assume that we split the functions $f_i$ into groups of sizes $m_i$. All in all, let us consider (5) instead of (1).

The problem (5) occurs in many applications, including distributed optimization and federated learning (Konečný et al., 2016; McMahan et al., 2017). In federated learning, many devices and machines (nodes) store local datasets that they do not share with other nodes. The local datasets are represented by functions $f_i$, and all nodes solve the common optimization problem (5). Due to

---

**Algorithm 2** PAGE with composition of samplings

---

1: **Input:** initial point $x^0 \in \mathbb{R}^d$, stepsize $\gamma > 0$, probability $p \in (0, 1]$, $g^0 = \nabla f(x^0)$
2: **for** $t = 0, 1, \ldots, T$ **do**
3:      $x^{t+1} = x^t - \gamma g^t$
4:      $c^{t+1} = \begin{cases} 1 & \text{with probability } p \\ 0 & \text{with probability } 1 - p \end{cases}$
5:      **if** $c^{t+1} = 1$ **then**
6:          $g^{t+1} = \nabla f(x^{t+1})$
         /* FL Interpretation: Calculate the full gradients $\nabla f_i$ on the nodes and collect them */
7:      **else**
8:          Generate samplings $\mathbf{S}_i^t$ for all $i \in [n]$
9:          $h_i^{t+1} = \mathbf{S}_i^t \left( \{ \nabla f_{ij}(x^{t+1}) - \nabla f_{ij}(x^t) \}_{j=1}^{m_i} \right)$ for all $i \in [n]$
         /* FL Interpretation: Calculate the mini-batches $h_i^{t+1}$ on the nodes */
10:          Generate a sampling $\mathbf{S}^t$ and set $g^{t+1} = g^t + \mathbf{S}^t \left( \{ h_i^{t+1} \}_{i=1}^n \right)$
         /* FL Interpretation: Collect $h_i^{t+1}$ only from the sampled nodes */
11:      **end if**
12: **end for**

---

privacy reasons and communication bottlenecks (Kairouz et al., 2021), it is infeasible to store and compute the functions $f_i$ locally in one machine.

In general, when we solve (1) in one machine, we have the freedom of choosing a sampling $\mathbf{S}$ for the functions $f_i$, which we have shown in Sec 3. However, in federated learning, a sampling of nodes or the functions $f_i$ is dictated by hardware limits or network quality (Kairouz et al., 2021). Still, each $i^{\text{th}}$ node can choose sampling $\mathbf{S}_i$ to sample the functions $f_{ij}$. As a result, we have a composition of the sampling $\mathbf{S}$ and the samplings $\mathbf{S}_i$ (see Algorithm 2).

**Assumption 6.** *For all $j \in [m_i], i \in [n]$, there exists a Lipschitz constant $L_{ij}$ such that $\|\nabla f_{ij}(x) - \nabla f_{ij}(y)\| \le L_{ij} \|x - y\|$ for all $x, y \in \mathbb{R}^d$.*

We now introduce the counterpart of Definitions 2 and 3.

**Definition 7.** *For all $i \in [n]$ and any sampling $\mathbf{S}_i \in \mathbb{S}(A_i, B_i, \{w_{ij}\}_{j=1}^{m_i})$, define constant $L_{i,+,w_i}$ such that*

$$\frac{1}{m_i} \sum_{j=1}^{m_i} \frac{1}{m_i w_{ij}} \|\nabla f_{ij}(x) - \nabla f_{ij}(y)\|^2 \le L_{i,+,w_i}^2 \|x - y\|^2 \quad \forall x, y \in \mathbb{R}^d.$$

**Definition 8.** *For all $i \in [n]$ and any sampling $\mathbf{S}_i \in \mathbb{S}(A_i, B_i, \{w_{ij}\}_{j=1}^{m_i})$, define constant $L_{i,\pm,w_i}$ such that*

$$\frac{1}{m_i} \sum_{j=1}^{m_i} \frac{1}{m_i w_{ij}} \|\nabla f_{ij}(x) - \nabla f_{ij}(y)\|^2 - \|\nabla f_i(x) - \nabla f_i(y)\|^2 \le L_{i,\pm,w_i}^2 \|x - y\|^2 \quad \forall x, y \in \mathbb{R}^d.$$

Let us provide the counterpart of Thm 5 for Algorithm 2.

**Theorem 9.** *Suppose that Assumptions 1, 2, 3, 6 hold and the samplings $\mathbf{S}^t \in \mathbb{S}(A, B, \{w_i\}_{i=1}^n)$ and the samplings $\mathbf{S}_i^t \in \mathbb{S}(A_i, B_i, \{w_{ij}\}_{j=1}^{m_i})$ for all $i \in [n]$. Moreover, $B \le 1$. Then Algorithm 2 has the convergence rate $\mathrm{E}\left[ \|\nabla f(\widehat{x}^T)\|^2 \right] \le \frac{2\Delta_0}{\gamma T}$, where*

$$\gamma \le \left( L_- + \sqrt{\frac{1-p}{p} \left( \frac{1}{n} \sum_{i=1}^n \left( \frac{A}{nw_i} + \frac{(1-B)}{n} \right) \left( (A_i - B_i) L_{i,+,w_i}^2 + B_i L_{i,\pm,w_i}^2 \right) + (A - B) L_{+,w}^2 + B L_{\pm,w}^2 \right)} \right)^{-1}.$$

The obtained theorem provides a general framework that helps analyze the convergence rates of the composition of samplings that satisfy Assumption 4. We discuss the obtained result in different contexts.

## 4.1 FEDERATED LEARNING

For simplicity, let us assume that the samplings $\mathbf{S}^t$ and $\mathbf{S}_i^t$ are *Uniform With Replacement* samplings with batch sizes $\tau$ and $\tau_i$ for all $i \in n$, accordingly, then to get $\varepsilon$-stationary point, it is enough to do $T := \Theta\left(\frac{\Delta_0}{\varepsilon}\left(L_- + \sqrt{\frac{1-p}{p\tau}\left(\frac{1}{n}\sum_{i=1}^n \frac{1}{\tau_i}L_{i,\pm}^2 + L_\pm^2\right)}\right)\right)$ iterations. Note that $T \geq \Theta\left(\frac{\Delta_0}{\varepsilon}\left(L_- + \sqrt{\frac{1-p}{p\tau}L_\pm^2}\right)\right)$ for all $\tau_i \geq 1$ for all $i \in [n]$. It means that after some point, there is no benefit in increasing batch sizes $\tau_i$. In order to balance $\frac{1}{n}\sum_{i=1}^n \frac{1}{\tau_i}L_{i,\pm}^2$ and $L_\pm^2$, one can take $\tau_i = \Theta\left(L_{i,\pm}^2/L_\pm^2\right)$. The constant $L_{i,\pm}^2$ captures the *intra-variance* inside $i^{\text{th}}$ node, while $L_\pm^2$ captures the *inter-variance* between nodes. If the *intra-variance* is small with respect to the *inter-variance*, then our theory suggests taking small batch sizes and vice versa.

## 4.2 *Stratified* SAMPLING

Let us provide another example that is closely related to (Zhao & Zhang, 2014). Let us consider (1) and use a variation of the *Stratified* sampling (Zhao & Zhang, 2014): we split the functions $f_i$ into $g = n/m$ groups, where $m$ is the number of functions in each group. Thus we get the problem (5) with $f(x) = \frac{1}{g}\sum_{i=1}^g \frac{1}{m}\sum_{j=1}^m f_{ij}(x)$. Let us assume that we always sample *all* groups, thus $A = B = 0$, and the sampling $\mathbf{S}_i^t$ are *Nice* samplings with batch sizes $\tau_1$ for all $i \in [n]$. Applying Thm 9, we get the convergence rate $T_{\text{group}} := \Theta\left(\frac{\Delta_0}{\varepsilon}\left(L_- + \sqrt{\frac{1-p}{pg\tau_1}\left(\frac{1}{g}\sum_{i=1}^g L_{i,\pm}^2\right)}\right)\right)$. At each iteration, the algorithm calculates $g\tau_1$ gradients, thus we should take $p = \frac{g\tau_1}{g\tau_1 + n}$ to get the complexity $N_{\text{group}} := \Theta\left(n + g\tau_1 T\right) = \Theta\left(n + \frac{\Delta_0}{\varepsilon}\left(g\tau_1 L_- + \sqrt{n}\sqrt{\frac{1}{g}\sum_{i=1}^g L_{i,\pm}^2}\right)\right)$. Let us take $\tau_1 \leq \max\left\{\frac{\sqrt{n}\sqrt{\frac{1}{g}\sum_{i=1}^g L_{i,\pm}^2}}{gL_-}, 1\right\}$ to obtain the complexity $N_{\text{group}} = \Theta\left(n + \frac{\Delta_0 \max\left\{\sqrt{n}\sqrt{\frac{1}{g}\sum_{i=1}^g L_{i,\pm}^2}, gL_-\right\}}{\varepsilon}\right)$. Comparing the complexity $N_{\text{group}}$ with the complexity $N_{\text{uniform}}$ from Sec 3, one can see that if split the functions $f_i$ in a "right way", such that $L_{i,\pm}$ is small for $i \in [n]$ (see Example 3), then we can get at least $\sqrt{n}/\sqrt{g}$ times improvement with the *Stratified* sampling.

## 5 EXPERIMENTS

We now provide experiments[1] with synthetic quadratic optimization tasks, where the functions $f_i$, in general, are nonconvex quadratic functions. Note that our goal here is to check whether the dependencies that our theory predicts are correct for the problem (1). The procedures that generate synthetic quadratic optimization tasks give us control over the choice of smoothness constants. All parameters, including the step sizes, are chosen as suggested by the corresponding theory. In the plots, we represent the relation between the norm of gradients and the number of gradient calculations $\nabla f_i$.

### 5.1 QUADRATIC OPTIMIZATION TASKS WITH VARIOUS HESSIAN VARIANCES $L_\pm$

Using Algorithm 3 (see Appendix), we generated various quadratic optimization tasks with different smoothness constants $L_\pm \in [0, 1.0]$ and fixed $L_- \approx 1.0$ (see Fig. 1). We choose $d = 10$, $n = 1000$, regularization $\lambda = 0.001$, and the noise scale $s \in \{0, 0.1, 0.5, 1\}$. According to Sec 3 and Table 2, the gradient complexity of original PAGE method ("Vanilla PAGE" in Fig. 1) is proportional to $L_+$. While the gradient complexity of the new analysis with the *Uniform With Replacement* sampling ("Uniform With Replacement" in Fig. 1) is proportional to $L_\pm$, which is always less or equal $L_+$. In Fig. 1, one can see that the smaller $L_\pm$ with respect to $L_+$, the better the performance of "Uniform With Replacement." Moreover, we provide experiments with the *Importance* sampling ("Importance" in Fig. 1) with $q_i = \frac{L_i}{\sum_{i=1}^n L_i}$ for all $i \in [n]$. This sampling has the best performance in all regimes.

---

[1]Our code: https://github.com/mysteryresearcher/sampling-in-optimal-sgd

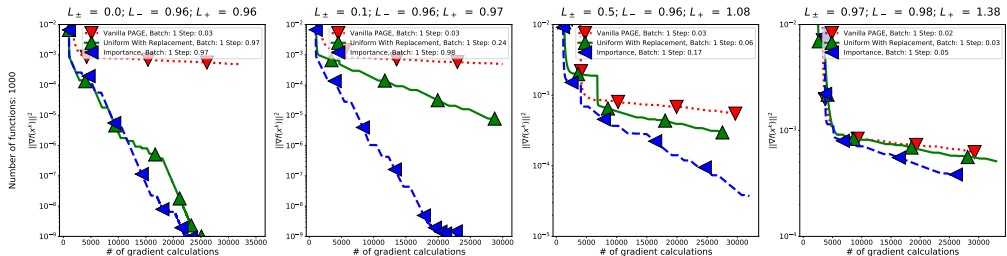

Figure 1: Comparison of samplings and methods on quadratic optimization tasks with various $L_\pm$.

## 5.2 QUADRATIC OPTIMIZATION TASKS WITH VARIOUS LOCAL LIPSCHITZ CONSTATNS $L_i$

Using Algorithm 4 (see Appendix), we synthesized various quadratic optimization tasks with different smoothness constants $L_i$ (see Fig. 2). We choose $d = 10$, $n = 1000$, the regularization $\lambda = 0.001$, and the noise scale $s \in \{0, 0.1, 0.5, 10.0\}$. We generated tasks in such way that the difference between $\max_i L_i$ and $\min_i L_i$ increases. First, one can see that the *Uniform With Replacement* sampling with the new analysis ("Uniform With Replacement" in Fig. 2) has better performance even in the cases of significant variations of $L_i$. Next, we see the stability of the *Importance* sampling ("Importance" in Fig. 2) with respect to this variations.

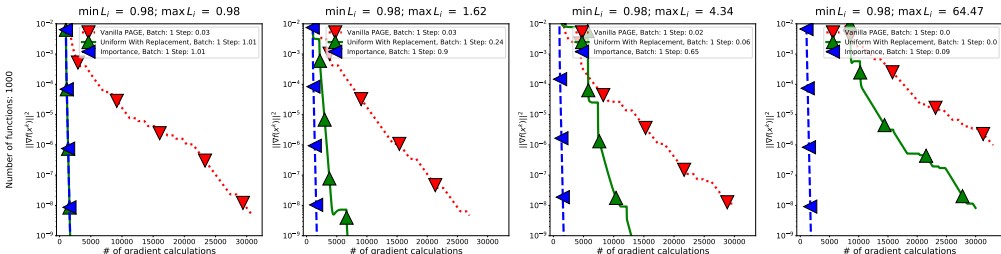

Figure 2: Comparison of samplings and methods on quadratic optimization tasks with various $L_i$.

## 5.3 NONCONVEX CLASSIFICATION PROBLEM WITH LIBSVM DATASETS

We now solve nonconvex machine learning tasks and compare samplings on LIBSVM datasets (Chang & Lin, 2011) (see details in Sec A.2). As in previous sections, PAGE with the *Importance* sampling performs better, especially in the *australian* dataset where the variation of $L_i$ is large.

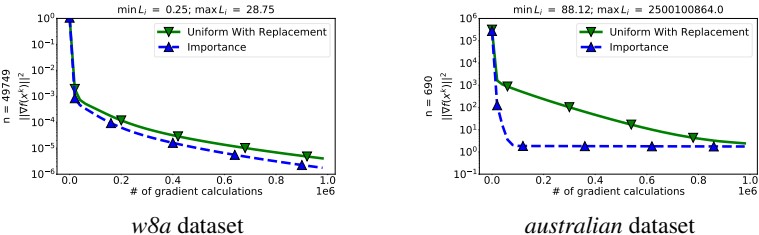

*w8a* dataset            *australian* dataset

Figure 3: Comparison of samplings on nonconvex machine learning tasks with LIBSVM datasets.

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

CONTENTS

# A EXTRA EXPERIMENTS AND DETAILS

## A.1 QUADRATIC OPTIMIZATION TASKS WITH VARIOUS BATCH SIZES.

In this section, we consider the same setup as in Sec 5.1. In Figure 4, we fix $L_\pm$, and show that the *Importance* sampling has better convergence rates with different batch sizes. Note that with large batches, the competitors reduce to the GD method, and the difference is not significant.

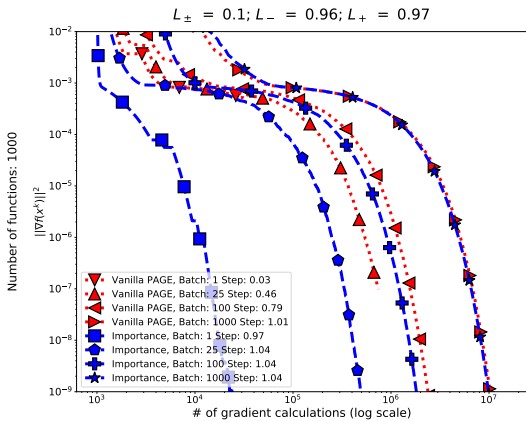

Figure 4: Comparison of samplings and methods with various batch sizes.

## A.2 DETAILS ON EXPERIMENTS WITH LIBSVM DATASETS

We compare the samplings on practical machine learnings with LIBSVM datasets (Chang & Lin, 2011) (under the 3-clause BSD license). Parameters of Algorithm 1 are chosen as suggested in Thm 5 and Cor 1. We take the parameters for *Uniform With Replacement* and *Importance* samplings from Table 1 with $q_i = \frac{L_i}{\sum_{i=1}^n L_i}$. We consider the logistic regression task with a nonconvex regularization (Wang et al., 2019)

$$f(x_1, x_2) := \frac{1}{n} \sum_{i=1}^n \left[ -\log\left( \frac{\exp\left(a_i^\top x_{y_i}\right)}{\sum_{y \in \{1,2\}} \exp\left(a_i^\top x_y\right)} \right) + \lambda \sum_{y \in \{1,2\}} \sum_{k=1}^d \frac{\{x_y\}_k^2}{1 + \{x_y\}_k^2} \right] \to \min_{x_1, x_2 \in \mathbb{R}^d},$$

where $\{\cdot\}_k$ is an indexing operation, $a_i \in \mathbb{R}^d$ is the feature of a $i^{\text{th}}$ sample, $y_i \in \{1, 2\}$ is the label of a $i^{\text{th}}$ sample, constant $\lambda = 0.001$. We fix batch size $\tau = 1$ and take *w8a* dataset (dimension $d = 300$, number of samples $n = 49,749$) and *australian* dataset (dimension $d = 14$, number of samples $n = 690$) from LIBSVM. For the logistic regression, the Lipschitz constants $L_i$ can be estimated. The distribution of Lipschitz constants $L_i$ across datapoints for that two datasets is presented in Fig. 5. We use Thm 4 to obtain $L_{+,w}^2$ and $L_{\pm,w}^2$.

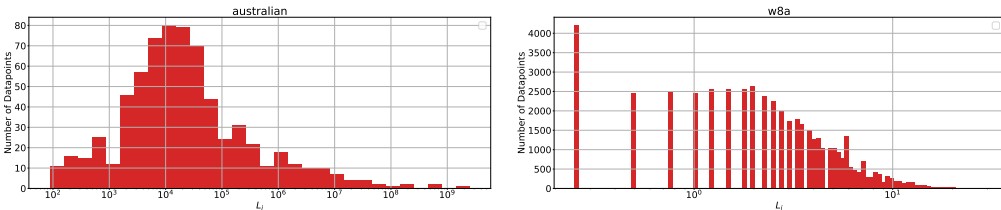

Figure 5: The distribution of Lipschitz constants $L_i$

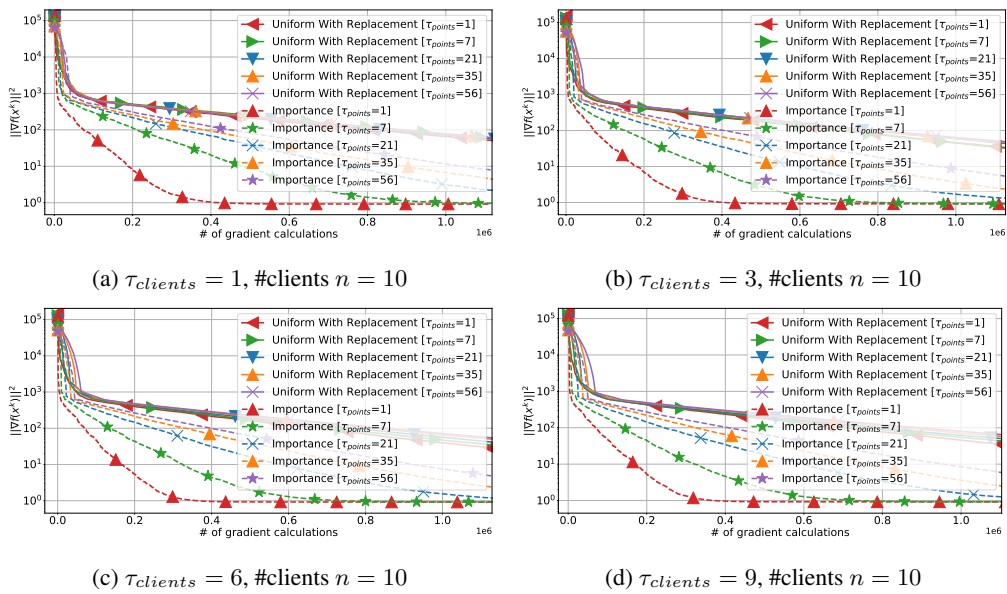

(a) $\tau_{clients} = 1$, #clients $n = 10$

(b) $\tau_{clients} = 3$, #clients $n = 10$

(c) $\tau_{clients} = 6$, #clients $n = 10$

(d) $\tau_{clients} = 9$, #clients $n = 10$

Figure 6: Comparison of methods on *australian* dataset from LIBSVM

### A.3 FEDERATED LEARNING EXPERIMENTS WITH LIBSVM DATASET

In this experiment[2], we compare the *Uniform With Replacement* sampling and the *Importance* sampling on the logistic regression task from Sec. A.2 in a distributed environment. The training of the models is carried on *australian* dataset from LIBSVM. The dataset is reshuffled with uniform distribution, and then it is split across $n = 10$ clients. In all experiments, we use Algorithm 2 with theoretical stepsizes according to Theorem 9. We take the parameters of the *Uniform With Replacement* and *Importance* samplings from Table 1 with $q_i = \frac{L_i}{\sum_{i=1}^{n} L_i}$.

According to Algorithm 2, we have the samplings $\mathbf{S}^t$ that sample clients, and the samplings $\mathbf{S}_i^t$ that sample data from the local datasets of clients. Algorithm 2 allows mixed sampling strategies that satisfy Assumption 4. For simplicity, we consider that the samplings $\mathbf{S}^t$ and $\mathbf{S}_i^t$ are of the same type.

For the logistic regression, the Lipschitz constants $L_i$ and $L_{ij}$ of the gradients of functions $f_i(x)$ and $f_{ij}(x)$ can be estimated. As in Sec A.2, we use Thm 4 to obtain the constants $L_{i,+,w}^2$, $L_{i,\pm,w}^2$, $L_{+,w}^2$ and $L_{\pm,w}^2$. The results of experiments are provided in Fig. 6. We denote by $\tau_{points}$ the batch size of the samplings $\mathbf{S}_i^t$ for all $i \in [n]$, and by $\tau_{clients}$ the batch size of the sampling $\mathbf{S}^t$. The number of gradient calculations in Fig. 6 stands for the total number of gradient calculations in all clients.

We demonstrate results for different values of the batch sizes $\tau_{clients}$ and $\tau_{points}$. As in previous experiments, the *Importance* sampling has better empirical performance than the *Uniform With Replacement* sampling. In addition to it, we observe that plots with small batch sizes $\tau_{points}$ converge faster.

### A.4 COMPUTING ENVIRONMENT

The code was written in Python 3.6.8 using PyTorch 1.9 (Paszke et al., 2019) and optimization research simulator FL_PyTorch (Burlachenko et al., 2021). The distributed environment was emulated on a machine with Intel(R) Xeon(R) Gold 6226R CPU @ 2.90GHz and 64 cores.

---

[2]Our code: https://github.com/mysteryresearcher/page_ab_fl_experiment_a3

## B  Auxiliary facts

We use the following auxiliary fact in out proofs:

1. Let us take a *random vector* $\xi \in \mathbb{R}^d$, then

$$\mathrm{E}\left[\|\xi\|^2\right] = \mathrm{E}\left[\|\xi - \mathrm{E}\left[\xi\right]\|^2\right] + \|\mathrm{E}\left[\xi\right]\|^2. \tag{6}$$

## C  Examples of Optimization Problems

**Example 1.** For simplicity, let us assume that $n$ is even. Let us consider the optimization problem (1) with $f_i(x) = \frac{a}{2}x^2 + \frac{b}{2}x^2$ for $i \in \{1, \cdots, n/2\}$ and $f_i(x) = -\frac{a}{2}x^2 + \frac{b}{2}x^2$ for $i \in \{n/2+1, \cdots, n\}$, where $x \in \mathbb{R}$ and $b \geq 0$. Then $f(x) = \frac{b}{2}x^2$ and

$$L_-^2 = \sup_{x \neq y} \frac{\|\nabla f(x) - \nabla f(y)\|^2}{\|x - y\|^2} = b^2.$$

Moreover,

$$L_+^2 = \sup_{x \neq y} \frac{\frac{1}{n}\sum_{i=1}^n \|\nabla f_i(x) - \nabla f_i(y)\|^2}{\|x - y\|^2} = \frac{1}{2}\left((a+b)^2 + (a-b)^2\right),$$

and we can take $a$ arbitrary large.

**Example 2.** Let us assume that $n \geq 2$ and consider the optimization problem (1) with $f_1(x) = \frac{b}{2}x^2$ and $f_i(x) = 0$ for $i \in \{2, \cdots, n\}$, where $x \in \mathbb{R}$ and $b \geq 0$. Then $f(x) = \frac{b}{2n}x^2$,

$$L_- = \sup_{x \neq y} \frac{\|\nabla f(x) - \nabla f(y)\|}{\|x - y\|} = \frac{b}{n},$$

$$\frac{1}{n}\sum_{i=1}^n L_i = \frac{1}{n}\sup_{x \neq y} \frac{\|\nabla f_1(x) - \nabla f_1(y)\|^2}{\|x - y\|^2} = \frac{b}{n},$$

and

$$L_+ = \sqrt{\sup_{x \neq y} \frac{\frac{1}{n}\sum_{i=1}^n \|\nabla f_i(x) - \nabla f_i(y)\|^2}{\|x - y\|^2}} = \frac{b}{\sqrt{n}}.$$

**Example 3.** Let us consider the optimization problem (5) with $f(x) = \frac{1}{g}\sum_{i=1}^g \frac{1}{m}\sum_{j=1}^m f_{ij}(x)$ and $f_{ij}(x) = \frac{b_i}{2}x^2$ for all $i \in [g]$ and $j \in [m]$, where $x \in \mathbb{R}$ and $b_1 \geq 0$ and $b_i = 0$ for all $i \in \{2, \ldots, g\}$. Then $f(x) = \frac{b_1}{2g}x^2$,

$$L_- = \sup_{x \neq y} \frac{\|\nabla f(x) - \nabla f(y)\|}{\|x - y\|} = \frac{b_1}{g},$$

$$L_{\pm}^2 = \sup_{x \neq y} \frac{\frac{1}{gm}\sum_{i=1}^g \sum_{j=1}^m \|\nabla f_{ij}(x) - \nabla f_{ij}(y)\|^2 - \|\nabla f(x) - \nabla f(y)\|^2}{\|x - y\|^2} = \left(\frac{1}{g} - \frac{1}{g^2}\right)b_1^2,$$

and

$$L_{i,\pm}^2 = \sup_{x \neq y} \frac{\frac{1}{m}\sum_{j=m}^n \|\nabla f_{ij}(x) - \nabla f_{ij}(y)\|^2 - \|\nabla f_i(x) - \nabla f_i(y)\|^2}{\|x - y\|^2} = 0 \quad \forall i \in [n].$$

Substituting the smoothness constants to the complexity $N_{\text{uniform}}$ from Sec 3 and $N_{\text{uniform}}$ from Sec 4, one can show that

$$N_{\text{uniform}} = \Theta\left(n + \frac{\Delta_0 \max\{\sqrt{n}L_{\pm}, L_-\}}{\varepsilon}\right) = \Theta\left(n + \frac{\Delta_0\sqrt{n}b_1}{\varepsilon\sqrt{g}}\right)$$

and

$$N_{\text{group}} = \Theta\left(n + \frac{\Delta_0 \max\left\{\sqrt{n}\sqrt{\frac{1}{g}\sum_{i=1}^g L_{i,\pm}^2}, gL_-\right\}}{\varepsilon}\right) = \Theta\left(n + \frac{\Delta_0 b_1}{\varepsilon}\right).$$

The complexity $N_{\text{group}}$ is $\sqrt{n}/\sqrt{g}$ times better than the complexity $N_{\text{uniform}}$.

## D  MISSING PROOFS

**Lemma 1.** *Suppose that Assumption 2 holds and let $x^{t+1} = x^t - \gamma g^t$. Then for any $g^t \in \mathbb{R}^d$ and $\gamma > 0$, we have*

$$f(x^{t+1}) \leq f(x^t) - \frac{\gamma}{2} \left\| \nabla f(x^t) \right\|^2 - \left( \frac{1}{2\gamma} - \frac{L_-}{2} \right) \left\| x^{t+1} - x^t \right\|^2 + \frac{\gamma}{2} \left\| g^t - \nabla f(x^t) \right\|^2. \quad (7)$$

*Proof.* Using Assumption 2, we have

$$f(x^{t+1}) \leq f(x^t) + \left\langle \nabla f(x^t), x^{t+1} - x^t \right\rangle + \frac{L_-}{2} \left\| x^{t+1} - x^t \right\|^2$$

$$= f(x^t) - \gamma \left\langle \nabla f(x^t), g^t \right\rangle + \frac{L_-}{2} \left\| x^{t+1} - x^t \right\|^2.$$

Next, due to $-\langle x, y \rangle = \frac{1}{2} \|x - y\|^2 - \frac{1}{2} \|x\|^2 - \frac{1}{2} \|y\|^2$, we obtain

$$f(x^{t+1}) \leq f(x^t) - \frac{\gamma}{2} \left\| \nabla f(x^t) \right\|^2 - \left( \frac{1}{2\gamma} - \frac{L_-}{2} \right) \left\| x^{t+1} - x^t \right\|^2 + \frac{\gamma}{2} \left\| g^t - \nabla f(x^t) \right\|^2.$$

$\square$

**Theorem 5.** *Suppose that Assumptions 1, 2, 3 hold and the samplings $\mathbf{S}^t \in \mathbb{S}(A, B, \{w_i\}_{i=1}^n)$. Then Algorithm 1 (PAGE) has the convergence rate $\mathrm{E}\left[ \left\| \nabla f(\widehat{x}^T) \right\|^2 \right] \leq \frac{2\Delta_0}{\gamma T}$, where $\gamma \leq \left( L_- + \sqrt{\frac{1-p}{p} \left( (A - B) L_{+,w}^2 + B L_{\pm,w}^2 \right)} \right)^{-1}$.*

*Proof.* We start with the estimation of the variance of the noise:

$$\mathrm{E}\left[ \left\| g^{t+1} - \nabla f(x^{t+1}) \right\|^2 \right]$$

$$= (1-p)\mathrm{E}\left[ \left\| g^t + \mathbf{S}^t \left( \{\nabla f_i(x^{t+1}) - \nabla f_i(x^t)\}_{i=1}^n \right) - \nabla f(x^{t+1}) \right\|^2 \right]$$

$$= (1-p) \left\| \mathbf{S}^t \left( \{\nabla f_i(x^{t+1}) - \nabla f_i(x^t)\}_{i=1}^n \right) - \left( \nabla f(x^{t+1}) - \nabla f(x^t) \right) \right\|^2 + (1-p) \left\| g^t - \nabla f(x^t) \right\|^2,$$

where we used the unbiasedness of the sampling. Using Assumption 4, we have

$$\mathrm{E}\left[ \left\| g^{t+1} - \nabla f(x^{t+1}) \right\|^2 \right]$$

$$\leq (1-p) \left( A \sum_{i=1}^n \frac{1}{n^2 w_i} \left\| \nabla f_i(x^{t+1}) - \nabla f_i(x^t) \right\|^2 - B \left\| \nabla f(x^{t+1}) - \nabla f(x^t) \right\|^2 \right)$$

$$+ (1-p) \left\| g^t - \nabla f(x^t) \right\|^2.$$

Using the definition of $L_{+,w}$ and $L_{\pm,w}$, we get

$$\mathrm{E}\left[ \left\| g^{t+1} - \nabla f(x^{t+1}) \right\|^2 \right]$$

$$\leq (1-p) \left( A \sum_{i=1}^n \frac{1}{n^2 w_i} \left\| \nabla f_i(x^{t+1}) - \nabla f_i(x^t) \right\|^2 - B \left\| \nabla f(x^{t+1}) - \nabla f(x^t) \right\|^2 \right)$$

$$+ (1-p) \left\| g^t - \nabla f(x^t) \right\|^2$$

$$= (1-p) \Bigg( (A - B) \left( \sum_{i=1}^n \frac{1}{n^2 w_i} \left\| \nabla f_i(x^{t+1}) - \nabla f_i(x^t) \right\|^2 \right) \qquad (8)$$

$$+ B \left( \sum_{i=1}^n \frac{1}{n^2 w_i} \left\| \nabla f_i(x^{t+1}) - \nabla f_i(x^t) \right\|^2 - \left\| \nabla f(x^{t+1}) - \nabla f(x^t) \right\|^2 \right) \Bigg)$$

$$+ (1-p) \left\| g^t - \nabla f(x^t) \right\|^2$$

$$\leq (1-p) \left( (A - B) L_{+,w}^2 + B L_{\pm,w}^2 \right) \left\| x^{t+1} - x^t \right\|^2 + (1-p) \left\| g^t - \nabla f(x^t) \right\|^2.$$

We now continue the proof using Lemma 1. We add (7) with $\frac{\gamma}{2p} \times$ (8), and take expectation to get

$$
\begin{aligned}
& \mathrm{E}\left[f(x^{t+1}) - f^* + \frac{\gamma}{2p} \left\|g^{t+1} - \nabla f(x^{t+1})\right\|^2\right] \\
& \leq \mathrm{E}\left[f\left(x^t\right) - f^* - \frac{\gamma}{2} \left\|\nabla f\left(x^t\right)\right\|^2 - \left(\frac{1}{2\gamma} - \frac{L_-}{2}\right) \left\|x^{t+1} - x^t\right\|^2 + \frac{\gamma}{2} \left\|g^t - \nabla f\left(x^t\right)\right\|^2\right] \\
& \quad + \frac{\gamma}{2p} \mathrm{E}\left[(1-p) \left\|g^t - \nabla f\left(x^t\right)\right\|^2 + (1-p)\left((A-B) L_{+,w}^2 + B L_{\pm,w}^2\right) \left\|x^{t+1} - x^t\right\|^2\right] \\
& = \mathrm{E}\left[f\left(x^t\right) - f^* + \frac{\gamma}{2p} \left\|g^t - \nabla f\left(x^t\right)\right\|^2 - \frac{\gamma}{2} \left\|\nabla f\left(x^t\right)\right\|^2\right. \\
& \quad \left. - \left(\frac{1}{2\gamma} - \frac{L_-}{2} - \frac{(1-p)\gamma}{2p}\left((A-B) L_{+,w}^2 + B L_{\pm,w}^2\right)\right) \left\|x^{t+1} - x^t\right\|^2\right] \\
& \leq \mathrm{E}\left[f\left(x^t\right) - f^* + \frac{\gamma}{2p} \left\|g^t - \nabla f\left(x^t\right)\right\|^2 - \frac{\gamma}{2} \left\|\nabla f\left(x^t\right)\right\|^2\right],
\end{aligned}
\tag{9}
$$

where the last inequality holds due to $\frac{1}{2\gamma} - \frac{L_-}{2} - \frac{(1-p)\gamma}{2p}\left((A-B) L_{+,w}^2 + B L_{\pm,w}^2\right) \geq 0$ by choosing stepsize

$$
\gamma \leq \left(L_- + \sqrt{\frac{1-p}{p}\left((A-B) L_{+,w}^2 + B L_{\pm,w}^2\right)}\right)^{-1}.
$$

Now, if we define $\Phi_t := f\left(x^t\right) - f^* + \frac{\gamma}{2p} \left\|g^t - \nabla f\left(x^t\right)\right\|^2$, then (9) can be written in the form

$$
\mathrm{E}\left[\Phi_{t+1}\right] \leq \mathrm{E}\left[\Phi_t\right] - \frac{\gamma}{2}\mathrm{E}\left[\left\|\nabla f\left(x^t\right)\right\|^2\right].
$$

Summing up from $t = 0$ to $T-1$, we get

$$
\mathrm{E}\left[\Phi_T\right] \leq \mathrm{E}\left[\Phi_0\right] - \frac{\gamma}{2}\sum_{t=0}^{T-1} \mathrm{E}\left[\left\|\nabla f\left(x^t\right)\right\|^2\right].
$$

Then according to the output of the algorithm, i.e., $\widehat{x}_T$ is randomly chosen from $\{x^t\}_{t\in[T]}$ and $\Phi_0 = f\left(x^0\right) - f^* + \frac{\gamma}{2p}\|g^0 - \nabla f\left(x^0\right)\|^2 = f\left(x^0\right) - f^* \stackrel{\text{def}}{=} \Delta_0$, we have

$$
\mathrm{E}\left[\left\|\nabla f\left(\widehat{x}_T\right)\right\|^2\right] \leq \frac{2\Delta_0}{\gamma T}.
$$

$\square$

**Theorem 6.** *Suppose that Assumptions 1, 2, 3, 5 and the samplings $\mathbf{S}^t \in \mathbb{S}(A, B, \{w_i\}_{i=1}^n)$.* *Then Algorithm 1* (PAGE) *has the convergence rate $\mathrm{E}\left[f(x^T)\right] - f^* \leq (1 - \gamma\mu)^T \Delta_0$, where* $\gamma \leq \min\left\{\left(L_- + \sqrt{\frac{2(1-p)}{p}\left((A-B) L_{+,w}^2 + B L_{\pm,w}^2\right)}\right)^{-1}, \frac{p}{2\mu}\right\}.$

*Proof.* From the proof of Thm 5, we know that

$$
\begin{aligned}
& \mathrm{E}\left[\left\|g^{t+1} - \nabla f(x^{t+1})\right\|^2\right] \\
& \leq (1-p)\left((A-B) L_{+,w}^2 + B L_{\pm,w}^2\right) \left\|x^{t+1} - x^t\right\|^2 + (1-p)\left\|g^t - \nabla f(x^t)\right\|^2.
\end{aligned}
\tag{10}
$$

Using Lemma 1, we add (7) with $\frac{\gamma}{p} \times$ (10), and take expectation to get

$$
\mathrm{E}\left[f(x^{t+1}) - f^* + \frac{\gamma}{p}\left\|g^{t+1} - \nabla f(x^{t+1})\right\|^2\right]
$$

$$
\leq \mathrm{E}\left[f\left(x^t\right) - f^* - \frac{\gamma}{2}\left\|\nabla f\left(x^t\right)\right\|^2 - \left(\frac{1}{2\gamma} - \frac{L_-}{2}\right)\left\|x^{t+1} - x^t\right\|^2 + \frac{\gamma}{2}\left\|g^t - \nabla f\left(x^t\right)\right\|^2\right]
$$

$$
+ \frac{\gamma}{p}\mathrm{E}\left[(1-p)\left\|g^t - \nabla f\left(x^t\right)\right\|^2 + (1-p)\left((A-B)L_{+,w}^2 + BL_{\pm,w}^2\right)\left\|x^{t+1} - x^t\right\|^2\right]
$$

$$
= \mathrm{E}\left[f\left(x^t\right) - f^* + \left(1 - \frac{p}{2}\right)\frac{\gamma}{p}\left\|g^t - \nabla f\left(x^t\right)\right\|^2 - \frac{\gamma}{2}\left\|\nabla f\left(x^t\right)\right\|^2\right.
$$

$$
\left. - \left(\frac{1}{2\gamma} - \frac{L_-}{2} - \frac{(1-p)\gamma}{p}\left((A-B)L_{+,w}^2 + BL_{\pm,w}^2\right)\right)\left\|x^{t+1} - x^t\right\|^2\right]
$$

$$
\leq \mathrm{E}\left[f\left(x^t\right) - f^* + \left(1 - \frac{p}{2}\right)\frac{\gamma}{p}\left\|g^t - \nabla f\left(x^t\right)\right\|^2 - \frac{\gamma}{2}\left\|\nabla f\left(x^t\right)\right\|^2\right],
$$

where the last inequality holds due to $\frac{1}{2\gamma} - \frac{L_-}{2} - \frac{(1-p)\gamma}{p}\left((A-B)L_{+,w}^2 + BL_{\pm,w}^2\right) \geq 0$ by choosing stepsize

$$
\gamma \leq \left(L_- + \sqrt{\frac{2(1-p)}{p}\left((A-B)L_{+,w}^2 + BL_{\pm,w}^2\right)}\right)^{-1}.
$$

Next, using Assumption 5 and $\gamma \leq \frac{p}{2\mu}$, we have

$$
\mathrm{E}\left[f(x^{t+1}) - f^* + \frac{\gamma}{p}\left\|g^{t+1} - \nabla f(x^{t+1})\right\|^2\right]
$$

$$
\leq (1 - \gamma\mu)\mathrm{E}\left[f\left(x^t\right) - f^* + \frac{\gamma}{p}\left\|g^t - \nabla f\left(x^t\right)\right\|^2\right].
$$

Unrolling the recursion and considering that $g^0 = \nabla f(x^0)$, we can complete the proof of theorem. $\square$

# E    DERIVATIONS OF THE PARAMETERS FOR THE SAMPLINGS

## E.1    *Nice* SAMPLING

Let $S$ be a random subset uniformly chosen from $[n]$ with a fixed cardinality $\tau$. Let us fix $a_1, \ldots, a_n \in \mathbb{R}^d$. A sampling $\mathbf{S}(a_1, \ldots, a_n) := \frac{1}{n}\sum_{i \in S}\frac{a_i}{p_i}$ is called the *Nice* sampling, where $p_i := \mathbf{Prob}(i \in S)$.

Let us bound $\mathrm{E}\left[\left\|\mathbf{S}(a_1, \ldots, a_n) - \frac{1}{n}\sum_{i=1}^n a_i\right\|^2\right]$ and find parameters from Assumption 4. Note that $|\mathbf{S}| = |S| = \tau$. We introduce auxiliary random variables

$$
\chi_i := \begin{cases} 1 & i \in S \\ 0 & \text{otherwise.} \end{cases}
$$

Due to $p_i = \mathbf{Prob}\,(i \in S) = \frac{\tau}{n}$, we have

$$
\begin{aligned}
\mathrm{E}\left[\left\|\frac{1}{n}\sum_{i \in S}\frac{a_i}{p_i}\right\|^2\right] &= \mathrm{E}\left[\left\|\frac{1}{\tau}\sum_{i=1}^{n}\chi_i a_i\right\|^2\right] \\
&= \frac{1}{\tau^2}\sum_{i=1}^{n}\mathrm{E}\left[\|\chi_i a_i\|^2\right] + \frac{1}{\tau^2}\sum_{i\neq j}\mathrm{E}\left[\langle\chi_i a_i, \chi_j a_j\rangle\right] \\
&= \frac{1}{\tau^2}\sum_{i=1}^{n}\mathrm{E}\left[\chi_i\right]\|a_i\|^2 + \frac{1}{\tau^2}\sum_{i\neq j}\mathrm{E}\left[\langle\chi_i, \chi_j\rangle\right]\langle a_i, a_j\rangle \\
&= \frac{1}{n\tau}\sum_{i=1}^{n}\|a_i\|^2 + \frac{\tau-1}{n(n-1)\tau}\sum_{i\neq j}\langle a_i, a_j\rangle \\
&= \frac{1}{n\tau}\sum_{i=1}^{n}\|a_i\|^2 + \frac{\tau-1}{n(n-1)\tau}\left(\left\|\sum_{i=1}^{n}a_i\right\|^2 - \sum_{i=1}^{n}\|a_i\|^2\right) \\
&= \frac{n-\tau}{\tau(n-1)}\frac{1}{n}\sum_{i=1}^{n}\|a_i\|^2 + \frac{\tau-1}{n(n-1)\tau}\left\|\sum_{i=1}^{n}a_i\right\|^2,
\end{aligned}
$$

where we use $\mathrm{E}\left[\chi_i^2\right] = \mathrm{E}\left[\chi_i\right] = \frac{\tau}{n}$ and $\mathrm{E}\left[\chi_i\chi_j\right] = \frac{\tau(\tau-1)}{n(n-1)}$, when $i \neq j$.

Finally, we have

$$
\begin{aligned}
\mathrm{E}\left[\left\|\frac{1}{n}\sum_{i \in S}\frac{a_i}{p_i} - \frac{1}{n}\sum_{i=1}^{n}a_i\right\|^2\right] &= \mathrm{E}\left[\left\|\frac{1}{n}\sum_{i \in S}\frac{a_i}{p_i}\right\|^2\right] - \left\|\frac{1}{n}\sum_{i=1}^{n}a_i\right\|^2 \\
&= \frac{n-\tau}{\tau(n-1)}\frac{1}{n}\sum_{i=1}^{n}\|a_i\|^2 + \frac{\tau-1}{n(n-1)\tau}\left\|\sum_{i=1}^{n}a_i\right\|^2 - \left\|\frac{1}{n}\sum_{i=1}^{n}a_i\right\|^2 \\
&= \frac{n-\tau}{\tau(n-1)}\left(\frac{1}{n}\sum_{i=1}^{n}\|a_i\|^2 - \left\|\frac{1}{n}\sum_{i=1}^{n}a_i\right\|^2\right).
\end{aligned}
$$

Thus we have $A = B = \frac{n-\tau}{\tau(n-1)}$ and $w_i = \frac{1}{n}$ for all $i \in [n]$.

### E.2  *Independent* SAMPLING

Let us define i.i.d. random variables

$$
\chi_i = \begin{cases} 1 & \text{with probability } p_i \\ 0 & \text{with probability } 1 - p_i, \end{cases}
$$

for all $i \in [n]$ and take $S := \{i \in [n]\,|\,\chi_i = 1\}$. We now fix $a_1, \ldots, a_n \in \mathbb{R}^d$. A sampling $\mathbf{S}(a_1, \ldots, a_n) := \frac{1}{n}\sum_{i \in S}\frac{a_i}{p_i}$ is called the *Independent* sampling, where $p_i := \mathbf{Prob}(i \in S)$.

We get

$$
\mathrm{E}\left[\left\|\frac{1}{n}\sum_{i\in S}\frac{a_i}{p_i}-\frac{1}{n}\sum_{i=1}^{n}a_i\right\|^2\right] = \mathrm{E}\left[\left\|\frac{1}{n}\sum_{i=1}^{n}\frac{1}{p_i}\chi_i a_i\right\|^2\right] - \left\|\frac{1}{n}\sum_{i=1}^{n}a_i\right\|^2
$$

$$
= \sum_{i=1}^{n}\frac{\mathrm{E}\left[\chi_i\right]}{n^2 p_i^2}\|a_i\|^2 + \sum_{i\neq j}\frac{\mathrm{E}\left[\chi_i\right]\mathrm{E}\left[\chi_j\right]}{n^2 p_i p_j}\langle a_i, a_j\rangle - \left\|\frac{1}{n}\sum_{i=1}^{n}a_i\right\|^2
$$

$$
= \sum_{i=1}^{n}\frac{1}{n^2 p_i}\|a_i\|^2 + \frac{1}{n^2}\left(\left\|\sum_{i=1}^{n}a_i\right\|^2 - \sum_{i=1}^{n}\|a_i\|^2\right) - \left\|\frac{1}{n}\sum_{i=1}^{n}a_i\right\|^2
$$

$$
= \frac{1}{n^2}\sum_{i=1}^{n}\left(\frac{1}{p_i}-1\right)\|a_i\|^2 .
$$

Thus we have $A = \frac{1}{\sum_{i=1}^{n}\frac{p_i}{1-p_i}}$, $B = 0$ and $w_i = \frac{\frac{p_i}{1-p_i}}{\sum_{i=1}^{n}\frac{p_i}{1-p_i}}$ for all $i \in [n]$.

### E.3  *Importance* AND *Uniform With Replacement* SAMPLING

Let us fix $\tau > 0$. For all $k \in [\tau]$, we define i.i.d. random variables

$$
\chi_k = \begin{cases} 1 & \text{with probability } q_1 \\ 2 & \text{with probability } q_2 \\ \quad\vdots \\ n & \text{with probability } q_n, \end{cases}
$$

where $(q_1, \ldots, q_n) \in \mathcal{S}^n$ (simple simplex). A sampling

$$
\mathbf{S}(a_1, \ldots, a_n) := \frac{1}{\tau}\sum_{k=1}^{\tau}\frac{a_{\chi_k}}{n q_{\chi_k}}
$$

is called the *Importance* sampling. The *Importance* sampling reduces to the *Uniform With Replacement* sampling when $q_i = 1/n$ for all $i \in [n]$. Note that $|\mathbf{S}| \leq \tau$.

Let us bound the variance

$$
\mathrm{E}\left[\left\|\frac{1}{\tau}\sum_{k=1}^{\tau}\frac{a_{\chi_k}}{n q_{\chi_k}}-\frac{1}{n}\sum_{i=1}^{n}a_i\right\|^2\right]
$$

$$
= \frac{1}{\tau^2}\sum_{k=1}^{\tau}\mathrm{E}\left[\left\|\frac{a_{\chi_k}}{n q_{\chi_k}}-\frac{1}{n}\sum_{i=1}^{n}a_i\right\|^2\right] + \frac{1}{\tau^2}\sum_{k\neq k'}\mathrm{E}\left[\left\langle\frac{a_{\chi_k}}{n q_{\chi_k}}-\frac{1}{n}\sum_{i=1}^{n}a_i, \frac{a_{\chi_{k'}}}{n q_{\chi_{k'}}}-\frac{1}{n}\sum_{i=1}^{n}a_i\right\rangle\right].
$$

Using the independents and unbiasedness of the random variables, the last term vanishes and we get

$$
\mathrm{E}\left[\left\|\frac{1}{\tau}\sum_{k=1}^{\tau}\frac{a_{\chi_k}}{n q_{\chi_k}}-\frac{1}{n}\sum_{i=1}^{n}a_i\right\|^2\right] = \frac{1}{\tau^2}\sum_{k=1}^{\tau}\mathrm{E}\left[\left\|\frac{a_{\chi_k}}{n q_{\chi_k}}-\frac{1}{n}\sum_{i=1}^{n}a_i\right\|^2\right]
$$

$$
\overset{(6)}{=} \frac{1}{\tau^2}\sum_{k=1}^{\tau}\mathrm{E}\left[\left\|\frac{a_{\chi_k}}{n q_{\chi_k}}\right\|^2\right] - \frac{1}{\tau}\left\|\frac{1}{n}\sum_{i=1}^{n}a_i\right\|^2
$$

$$
= \frac{1}{\tau}\sum_{i=1}^{n}q_i\left\|\frac{a_i}{n q_i}\right\|^2 - \frac{1}{\tau}\left\|\frac{1}{n}\sum_{i=1}^{n}a_i\right\|^2
$$

$$
= \frac{1}{\tau}\left(\frac{1}{n}\sum_{i=1}^{n}\frac{1}{n q_i}\|a_i\|^2 - \left\|\frac{1}{n}\sum_{i=1}^{n}a_i\right\|^2\right).
$$

Thus we have $A = B = \frac{1}{\tau}$, and $w_i = q_i$ for all $i \in [n]$.

### E.4 *Extended Nice* SAMPLING

In this section, we analyze the extension of *Nice* sampling. First, we $l_i$ times repeat each vector $a_i$, then we use the *Nice* sampling. We define

$$
\tilde{a}_i := \begin{cases}
\frac{\sum_{j=1}^{n} l_j}{nl_1} a_1 & 1 \le i \le l_1 \\
\frac{\sum_{j=1}^{n} l_j}{nl_2} a_2 & l_1 + 1 \le i \le l_1 + l_2 \\
\qquad\vdots \\
\frac{\sum_{j=1}^{n} l_j}{nl_n} a_n & \sum_{j=1}^{n-1} l_j \le i \le \sum_{j=1}^{n} l_j,
\end{cases}
$$

where $a_i \in \mathbb{R}^d$ and $l_i \ge 1$ for all $i \in [n]$. Then we have

$$
\frac{1}{n} \sum_{i=1}^{n} a_i(x) = \frac{1}{N} \sum_{i=1}^{N} \tilde{a}_i(x),
$$

where $N := \sum_{j=1}^{n} l_j$. Also, we denote $N_k := \sum_{j=1}^{k} l_j$.

For some $\tau > 0$, we apply the *Nice* sampling method:

$$
\mathbf{S}(a_1, \dots, a_n) := \frac{1}{N} \sum_{i \in S} \frac{\tilde{a}_i}{p_i} = \sum_{i=1}^{N} \frac{1}{\tau} \chi_i \tilde{a}_i,
$$

where

$$
\chi_i = \begin{cases} 1 & i \in S \\ 0 & \text{otherwise} \end{cases}, \quad p_i = \mathbf{Prob}\,(i \in S),
$$

and $S$ is a random set with cardinality $\tau$ from $[N]$. The sampling $\mathbf{S}(a_1, \dots, a_n)$ is called the *Extended Nice* sampling.

We now ready to bound the variance. Using the results for the *Nice* sampling, we obtain

$$
\mathbb{E}\left[\left\| \mathbf{S}(a_1, \dots, a_n) - \frac{1}{n} \sum_{i=1}^{n} a_i(x) \right\|^2\right]
$$

$$
= \mathbb{E}\left[\left\| \mathbf{S}(a_1, \dots, a_n) - \frac{1}{N} \sum_{i=1}^{N} \tilde{a}_i(x) \right\|^2\right]
$$

$$
= \frac{n-\tau}{\tau(n-1)} \frac{1}{N} \sum_{i=1}^{N} \|\tilde{a}_i\|^2 - \frac{n-\tau}{\tau(n-1)} \left\| \frac{1}{N} \sum_{i=1}^{N} \tilde{a}_i \right\|^2
$$

$$
= \frac{n-\tau}{\tau(n-1)} \left( \frac{1}{N} \left(\frac{N}{nl_1}\right)^2 \sum_{i=1}^{N_1} \|a_1\|^2 + \frac{1}{N} \left(\frac{N}{nl_2}\right)^2 \sum_{i=N_1+1}^{N_2} \|a_2\|^2 \right.
$$

$$
\left. + \cdots + \frac{1}{N} \left(\frac{N}{nl_n}\right)^2 \sum_{i=N_{n-1}+1}^{N} \|a_n\|^2 \right) - \frac{n-\tau}{\tau(n-1)} \left\| \frac{1}{n} \sum_{i=1}^{n} a_i \right\|^2
$$

$$
= \frac{n-\tau}{\tau(n-1)} \left( \frac{N}{nl_1} \frac{1}{n} \|a_1\|^2 + \frac{N}{nl_2} \frac{1}{n} \|a_2\|^2 + \cdots + \frac{N}{nl_n} \frac{1}{n} \|a_n\|^2 \right) - \frac{n-\tau}{\tau(n-1)} \left\| \frac{1}{n} \sum_{i=1}^{n} a_i \right\|^2
$$

$$
= \frac{n-\tau}{\tau(n-1)} \left( \sum_{i=1}^{n} \frac{1}{n^2 w_i} \|a_i\|^2 \right) - \frac{n-\tau}{\tau(n-1)} \left\| \frac{1}{n} \sum_{i=1}^{n} a_i \right\|^2
$$

where $w_i = \frac{l_i}{N}$. Thus we have $A = B = \frac{n-\tau}{\tau(n-1)}$ and $w_i = \frac{l_i}{N}$ for $i \in [n]$.

## F    THE OPTIMAL CHOICE OF $w_i$

Let us consider $L_{+,w}^2$ and $L_{\pm,w}^2$. In Sec 2, we show that one can take $L_{+,w}^2 = L_{\pm,w}^2 = \frac{1}{n}\sum_{i=1}^n \frac{1}{nw_i}L_i^2$. Let us minimize $\frac{1}{n}\sum_{i=1}^n \frac{1}{nw_i}L_i^2$ with respect to the weights $w_i$ such that $w_1,\ldots,w_n \geq 0$ and $\sum_{i=1}^n w_i = 1$. Using the method of Lagrange multipliers, we can construct a Lagrangian

$$\mathcal{L}(w,\lambda) := \frac{1}{n}\sum_{i=1}^n \frac{1}{nw_i}L_i^2 - \lambda\left(\sum_{i=1}^n w_i - 1\right).$$

Next, we calculate partial derivatives

$$\frac{\partial \mathcal{L}}{\partial w_i} = -\frac{1}{n^2 w_i^2}L_i^2 - \lambda = 0 \forall i \in [n]$$

and get

$$w_i^2 = -\frac{L_i^2}{n^2\lambda}.$$

Using $\sum_{i=1}^n w_i = 1$, we can show that the weights $w_i^* = \frac{L_i}{\sum_{i=1}^n L_i}$ are the solutions of the minimization problem. Moreover,

$$L_{\pm,w^*}^2 = \frac{1}{n}\sum_{i=1}^n \frac{1}{nw_i^*}L_i^2 = \left(\frac{1}{n}\sum_{i=1}^n L_i\right)^2.$$

## G    THE COMPLEXITY OF ALGORITHM 1 WITH THE *Importance* SAMPLING

The expected number of gradient calculations $\nabla f_i$ of Algorithm 1 with the *Importance* sampling, the optimal $w_i^*$ from Sec. F, and $\tau \leq \max\left\{\frac{\sqrt{n}L_{\pm,w}}{L_-}, 1\right\}$ equals

$$N_{\text{importance}} = \mathcal{O}\left(n + \frac{\Delta_0}{\varepsilon}\tau\left(L_- + \frac{\sqrt{n}}{\tau}\sqrt{\frac{1}{n}\sum_{i=1}^n \frac{1}{nw_i^*}L_i^2}\right)\right)$$

$$= \mathcal{O}\left(n + \frac{\Delta_0}{\varepsilon}\tau\left(L_- + \frac{\sqrt{n}}{\tau}\frac{1}{n}\sum_{i=1}^n L_i\right)\right)$$

$$= \mathcal{O}\left(n + \frac{\Delta_0}{\varepsilon}\sqrt{n}L_{\pm,w^*} + \frac{\Delta_0\sqrt{n}\left(\frac{1}{n}\sum_{i=1}^n L_i\right)}{\varepsilon}\right)$$

$$= \mathcal{O}\left(n + \frac{\Delta_0\sqrt{n}\left(\frac{1}{n}\sum_{i=1}^n L_i\right)}{\varepsilon}\right).$$

## H    MISSING PROOFS: THE COMPOSITION OF SAMPLINGS

**Lemma 2.** *Let us assume that a random sampling function* $\mathbf{S}$ *satisfies Assumption 4 with some* $A, B$ *and weights* $w_i$, *and a random sampling function* $\mathbf{S}_i$ *satisfy Assumption 4 with some* $A_i, B_i$ *and weights* $w_{ij}$ *for all* $i \in [n]$. *Moreover,* $B \leq 1$. *Then*

$$\mathrm{E}\left[\left\|\mathbf{S}\left(\mathbf{S}_1\left(a_{11},\ldots,a_{1m_1}\right),\ldots,\mathbf{S}_n\left(a_{n1},\ldots,a_{nm_n}\right)\right) - \frac{1}{n}\sum_{i=1}^n\left(\frac{1}{m_i}\sum_{j=1}^{m_i}a_{ij}\right)\right\|^2\right]$$

$$\leq \frac{1}{n}\sum_{i=1}^n\left(\frac{A}{nw_i} + \frac{(1-B)}{n}\right)\left(\frac{A_i}{m_i}\sum_{j=1}^{m_i}\frac{1}{m_i w_{ij}}\|a_{ij}\|^2 - B_i\left\|\frac{1}{m_i}\sum_{j=1}^{m_i}a_{ij}\right\|^2\right)$$

$$+ \frac{A}{n} \sum_{i=1}^{n} \frac{1}{nw_i} \left\| \frac{1}{m_i} \sum_{j=1}^{m_i} a_{ij} \right\|^2 - B \left\| \frac{1}{n} \sum_{i=1}^{n} \left( \frac{1}{m_i} \sum_{j=1}^{m_i} a_{ij} \right) \right\|^2,$$

where $a_{ij} \in \mathbb{R}^d$ for all $j \in [m_i]$ and $i \in [n]$.

*Proof.* We denote $\widehat{a}_i := \mathbf{S}_i (a_{i1}, \dots, a_{im_i})$ and $a_i := \frac{1}{m_i} \sum_{j=1}^{m_i} a_{ij}$. Using (6), we have

$$\mathrm{E} \left[ \left\| \mathbf{S} (\widehat{a}_1, \dots, \widehat{a}_n) - \frac{1}{n} \sum_{i=1}^{n} a_i \right\|^2 \right]$$

$$= \mathrm{E} \left[ \mathrm{E}_S \left[ \left\| \mathbf{S} (\widehat{a}_1, \dots, \widehat{a}_n) - \frac{1}{n} \sum_{i=1}^{n} a_i \right\|^2 \right] \right]$$

$$= \mathrm{E} \left[ \mathrm{E}_S \left[ \left\| \mathbf{S} (\widehat{a}_1, \dots, \widehat{a}_n) - \frac{1}{n} \sum_{i=1}^{n} \widehat{a}_i \right\|^2 \right] \right] + \mathrm{E} \left[ \left\| \frac{1}{n} \sum_{i=1}^{n} \widehat{a}_i - \frac{1}{n} \sum_{i=1}^{n} a_i \right\|^2 \right].$$

Next, using Assumption 4 for the sampling $\mathbf{S}$, we get

$$\mathrm{E} \left[ \left\| \mathbf{S} (\widehat{a}_1, \dots, \widehat{a}_n) - \frac{1}{n} \sum_{i=1}^{n} a_i \right\|^2 \right]$$

$$\leq A \frac{1}{n} \sum_{i=1}^{n} \frac{1}{nw_i} \mathrm{E} \left[ \|\widehat{a}_i\|^2 \right] - B \mathrm{E} \left[ \left\| \frac{1}{n} \sum_{i=1}^{n} \widehat{a}_i \right\|^2 \right]$$

$$+ \mathrm{E} \left[ \left\| \frac{1}{n} \sum_{i=1}^{n} \widehat{a}_i - \frac{1}{n} \sum_{i=1}^{n} a_i \right\|^2 \right].$$

Due to (6), we obtain

$$\mathrm{E} \left[ \left\| \mathbf{S} (\widehat{a}_1, \dots, \widehat{a}_n) - \frac{1}{n} \sum_{i=1}^{n} a_i \right\|^2 \right]$$

$$\leq A \frac{1}{n} \sum_{i=1}^{n} \frac{1}{nw_i} \mathrm{E} \left[ \|\widehat{a}_i - a_i\|^2 \right] + A \frac{1}{n} \sum_{i=1}^{n} \frac{1}{nw_i} \|a_i\|^2$$

$$- B \mathrm{E} \left[ \left\| \frac{1}{n} \sum_{i=1}^{n} \widehat{a}_i - \frac{1}{n} \sum_{i=1}^{n} a_i \right\|^2 \right] - B \left\| \frac{1}{n} \sum_{i=1}^{n} a_i \right\|^2$$

$$+ \mathrm{E} \left[ \left\| \frac{1}{n} \sum_{i=1}^{n} \widehat{a}_i - \frac{1}{n} \sum_{i=1}^{n} a_i \right\|^2 \right]$$

$$= A \frac{1}{n} \sum_{i=1}^{n} \frac{1}{nw_i} \mathrm{E} \left[ \|\widehat{a}_i - a_i\|^2 \right] + A \frac{1}{n} \sum_{i=1}^{n} \frac{1}{nw_i} \|a_i\|^2$$

$$+ (1 - B) \mathrm{E} \left[ \left\| \frac{1}{n} \sum_{i=1}^{n} \widehat{a}_i - \frac{1}{n} \sum_{i=1}^{n} a_i \right\|^2 \right] - B \left\| \frac{1}{n} \sum_{i=1}^{n} a_i \right\|^2$$

$$= A \frac{1}{n} \sum_{i=1}^{n} \frac{1}{nw_i} \mathrm{E} \left[ \|\widehat{a}_i - a_i\|^2 \right] + A \frac{1}{n} \sum_{i=1}^{n} \frac{1}{nw_i} \|a_i\|^2$$

$$+ \frac{(1 - B)}{n^2} \sum_{i=1}^{n} \mathrm{E} \left[ \|\widehat{a}_i - a_i\|^2 \right] - B \left\| \frac{1}{n} \sum_{i=1}^{n} a_i \right\|^2$$

$$= \frac{1}{n} \sum_{i=1}^{n} \left( \frac{A}{nw_i} + \frac{(1-B)}{n} \right) \mathrm{E} \left[ \|\widehat{a}_i - a_i\|^2 \right] + A \frac{1}{n} \sum_{i=1}^{n} \frac{1}{nw_i} \|a_i\|^2 - B \left\| \frac{1}{n} \sum_{i=1}^{n} a_i \right\|^2.$$

Using Assumption 4 for the samplings $\mathbf{S}_i$, we have

$$\mathrm{E} \left[ \left\| \mathbf{S}(\widehat{a}_1, \ldots, \widehat{a}_n) - \frac{1}{n} \sum_{i=1}^{n} a_i \right\|^2 \right]$$

$$\leq \frac{1}{n} \sum_{i=1}^{n} \left( \frac{A}{nw_i} + \frac{(1-B)}{n} \right) \left( A_i \frac{1}{m_i} \sum_{j=1}^{m_i} \frac{1}{m_i w_{ij}} \|a_{ij}\|^2 - B_i \|a_i\|^2 \right)$$

$$+ A \frac{1}{n} \sum_{i=1}^{n} \frac{1}{nw_i} \|a_i\|^2 - B \left\| \frac{1}{n} \sum_{i=1}^{n} a_i \right\|^2.$$

$\square$

**Theorem 9.** *Suppose that Assumptions 1, 2, 3, 6 hold and the samplings $\mathbf{S}^t \in \mathbb{S}(A, B, \{w_i\}_{i=1}^{n})$ and the samplings $\mathbf{S}_i^t \in \mathbb{S}(A_i, B_i, \{w_{ij}\}_{j=1}^{m_i})$ for all $i \in [n]$. Moreover, $B \leq 1$. Then Algorithm 2 has the convergence rate $\mathrm{E} \left[ \|\nabla f(\widehat{x}^T)\|^2 \right] \leq \frac{2\Delta_0}{\gamma T}$, where*

$$\gamma \leq \left( L_- + \sqrt{ \frac{1-p}{p} \left( \frac{1}{n} \sum_{i=1}^{n} \left( \frac{A}{nw_i} + \frac{(1-B)}{n} \right) \left( (A_i - B_i) L_{i,+,w_i}^2 + B_i L_{i,\pm,w_i}^2 \right) + (A-B) L_{+,w}^2 + B L_{\pm,w}^2 \right) } \right)^{-1}.$$

*Proof.* We start with the estimation of the variance of the noise:

$$\mathrm{E} \left[ \|g^{t+1} - \nabla f(x^{t+1})\|^2 \right]$$

$$= (1-p) \mathrm{E} \left[ \left\| g^t + \mathbf{S} \left( \{ \mathbf{S}_i ( \{ \nabla f_{ij}(x^{t+1}) - \nabla f_{ij}(x^t) \}_{j=1}^{m_i} ) \}_{i=1}^{n} \right) - \nabla f(x^{t+1}) \right\|^2 \right]$$

$$= (1-p) \left\| \mathbf{S} \left( \{ \mathbf{S}_i ( \{ \nabla f_{ij}(x^{t+1}) - \nabla f_{ij}(x^t) \}_{j=1}^{m_i} ) \}_{i=1}^{n} \right) - \left( \nabla f(x^{t+1}) - \nabla f(x^t) \right) \right\|^2 + (1-p) \left\| g^t - \nabla f(x^t) \right\|^2,$$

where we used the unbiasedness of the composition of samplings. Using Lemma 2, we have

$$\mathrm{E} \left[ \|g^{t+1} - \nabla f(x^{t+1})\|^2 \right]$$

$$\leq (1-p) \left( \frac{1}{n} \sum_{i=1}^{n} \left( \frac{A}{nw_i} + \frac{(1-B)}{n} \right) \left( \frac{A_i}{m_i} \sum_{j=1}^{m_i} \frac{1}{m_i w_{ij}} \|\nabla f_{ij}(x^{t+1}) - \nabla f_{ij}(x^t)\|^2 - B_i \|\nabla f_i(x^{t+1}) - \nabla f_i(x^t)\|^2 \right) \right.$$

$$\left. + \frac{A}{n} \sum_{i=1}^{n} \frac{1}{nw_i} \|\nabla f_i(x^{t+1}) - \nabla f_i(x^t)\|^2 - B \|\nabla f(x^{t+1}) - \nabla f(x^t)\|^2 \right)$$

$$+ (1-p) \left\| g^t - \nabla f(x^t) \right\|^2.$$

Using Definitions 2, 3, 7 and 8, we get

$$\mathrm{E} \left[ \|g^{t+1} - \nabla f(x^{t+1})\|^2 \right]$$

$$\leq (1-p) \left( \frac{1}{n} \sum_{i=1}^{n} \left( \frac{A}{nw_i} + \frac{(1-B)}{n} \right) \left( \frac{A_i - B_i}{m_i} \sum_{j=1}^{m_i} \frac{1}{m_i w_{ij}} \|\nabla f_{ij}(x^{t+1}) - \nabla f_{ij}(x^t)\|^2 \right. \right.$$

$$\left. \left. + B_i \left( \frac{1}{m_i} \sum_{j=1}^{m_i} \frac{1}{m_i w_{ij}} \|\nabla f_{ij}(x^{t+1}) - \nabla f_{ij}(x^t)\|^2 - \|\nabla f_i(x^{t+1}) - \nabla f_i(x^t)\|^2 \right) \right) \right)$$

$$+ \frac{A - B}{n} \sum_{i=1}^{n} \frac{1}{nw_i} \|\nabla f_i(x^{t+1}) - \nabla f_i(x^t)\|^2$$

$$+ B \left( \frac{1}{n} \sum_{i=1}^{n} \frac{1}{nw_i} \left\| \nabla f_i(x^{t+1}) - \nabla f_i(x^t) \right\|^2 - \left\| \nabla f(x^{t+1}) - \nabla f(x^t) \right\|^2 \right) \right)$$

$$+ (1-p) \left\| g^t - \nabla f(x^t) \right\|^2$$

$$\leq (1-p) \left( \frac{1}{n} \sum_{i=1}^{n} \left( \frac{A}{nw_i} + \frac{(1-B)}{n} \right) \left( (A_i - B_i) L_{i,+,w_i}^2 + B_i L_{i,\pm,w_i}^2 \right) + (A-B) L_{+,w}^2 + B L_{\pm,w}^2 \right) \left\| x^{t+1} - x^t \right\|^2$$

$$+ (1-p) \left\| g^t - \nabla f(x^t) \right\|^2 .$$

From this point the proof of theorem repeats the proof of Thm 5 with

$$\frac{1}{n} \sum_{i=1}^{n} \left( \frac{A}{nw_i} + \frac{(1-B)}{n} \right) \left( (A_i - B_i) L_{i,+,w_i}^2 + B_i L_{i,\pm,w_i}^2 \right) + (A-B) L_{+,w}^2 + B L_{\pm,w}^2$$

instead of

$$(A-B) L_{+,w}^2 + B L_{\pm,w}^2 .$$

$\square$

## I  ARTIFICIAL QUADRATIC OPTIMIZATION TASKS

In this section, we provide algorithms that we use to generate artificial optimization tasks for experiments. Algorithm 3 and Algorithm 4 allow us to control the smoothness constants $L_\pm$ and $L_i$, accordingly, via the noise scales.

---

**Algorithm 3** Generate quadratic optimization task with controlled $L_\pm$ (homogeneity)

---

1: **Parameters:** number nodes $n$, dimension $d$, regularizer $\lambda$, and noise scale $s$.
2: **for** $i = 1, \ldots, n$ **do**
3:      Generate random noises $\nu_i^s = 1 + s\xi_i^s$ and $\nu_i^b = s\xi_i^b$, i.i.d. $\xi_i^s, \xi_i^b \sim \text{NormalDistribution}(0,1)$
4:      Take vector $b_i = \frac{\nu_i^s}{4}(-1 + \nu_i^b, 0, \cdots, 0) \in \mathbb{R}^d$
5:      Take the initial tridiagonal matrix

$$
\mathbf{A}_i = \frac{\nu_i^s}{4}
\begin{pmatrix}
2 & -1 & & 0 \\
-1 & \ddots & \ddots & \\
& \ddots & \ddots & -1 \\
0 & & -1 & 2
\end{pmatrix}
\in \mathbb{R}^{d \times d}
$$

6: **end for**
7: Take the mean of matrices $\mathbf{A} = \frac{1}{n} \sum_{i=1}^{n} \mathbf{A}_i$
8: Find the minimum eigenvalue $\lambda_{\min}(\mathbf{A})$
9: **for** $i = 1, \ldots, n$ **do**
10:      Update matrix $\mathbf{A}_i = \mathbf{A}_i + (\lambda - \lambda_{\min}(\mathbf{A}))\mathbf{I}$
11: **end for**
12: Take starting point $x^0 = (\sqrt{d}, 0, \cdots, 0)$
13: **Output:** matrices $\mathbf{A}_1, \cdots, \mathbf{A}_n$, vectors $b_1, \cdots, b_n$, starting point $x^0$

---

---

**Algorithm 4** Generate quadratic optimization task with controlled $L_i$

---

1: **Parameters:** number nodes $n$, dimension $d$, regularizer $\lambda$, and noise scale $s$.
2: **for** $i = 1, \ldots, n$ **do**
3:      Generate random noises $\nu_i^s = 1 + s\xi_i^s$, where i.i.d. $\xi_i^s \sim \text{ExponentialfDistribution}(1.0)$
4:      Generate random noises $\nu_i^b = s\xi_i^b$, i.i.d. $\xi_i^b \sim \text{NormalDistribution}(0,1)$
5:      Take vector $b_i = (-\frac{1}{4} + \nu_i^b, 0, \cdots, 0) \in \mathbb{R}^d$
6:      Take the initial tridiagonal matrix

$$
\mathbf{A}_i = \frac{\nu_i^s}{4}
\begin{pmatrix}
2 & -1 & & 0 \\
-1 & \ddots & \ddots & \\
& \ddots & \ddots & -1 \\
0 & & -1 & 2
\end{pmatrix}
\in \mathbb{R}^{d \times d}
$$

7: **end for**
8: Take starting point $x^0 = (\sqrt{d}, 0, \cdots, 0)$
9: **Output:** matrices $\mathbf{A}_1, \cdots, \mathbf{A}_n$, vectors $b_1, \cdots, b_n$, starting point $x^0$

---

