# OpenReview forum: "Sharper Rates and Flexible Framework for Nonconvex SGD with Client and Data Sampling"
_ICLR.cc/2023/Conference — Submitted to ICLR 2023_

### Official Review · Reviewer_paK1 · 2022-10-23

**Confidence:** 4
**Correctness:** 3
**Technical Novelty And Significance:** 1
**Empirical Novelty And Significance:** Not applicable
**Recommendation:** 3

**Clarity, Quality, Novelty And Reproducibility:**

The clarity of this paper is proper. However, its quality and originality are quite limited due to incremental contributions; see also my detailed comments below.

**Strength And Weaknesses:**

The new sampling assumption is justified by considering several representative sampling schemes. However, the technique is a direct combination of (Li et al., 2021) and (Szlendak et al., 2021), rendering this manuscript far from possible publication in ICLR.


[Li et al., 2021] PAGE: A simple and optimal probabilistic gradient estimator for nonconvex optimization. In International Conference on
Machine Learning, pp. 6286–6295. PMLR, 2021.

[Szlendak et al., 2021] Permutation compressors for provably faster distributed nonconvex optimization. arXiv preprint arXiv:2110.03300, 2021.

**Summary Of The Paper:**

This paper considers a probabilistic gradient method (PAGE). The authors invoke different kinds of sampling schemes and claim that the new result is shaper than the original one.

**Summary Of The Review:**

The techniques are copies of those of (Li et al., 2021) and (Szlendak et al., 2021). I obtain this conclusion by reading their proof of Theorem 5 and that of Thm 1 in (Li et al., 2021) and that of Thm 4 in (Szlendak et al., 2021), which is a step-by-step copy-style. I would say the essential contribution is that the authors changed the compressor's property to a sampling property. The authors verified that several sampling schemes do satisfy their Assumption 4 (Indeed, an unbiased compressor seems similar to unbiased sampling in terms of analysis). It is good to know the result. However, I have to say that such a contribution does not meet the requirement for possible publication in ICLR.

[Li et al., 2021] PAGE: A simple and optimal probabilistic gradient estimator for nonconvex optimization. In International Conference on
Machine Learning, pp. 6286–6295. PMLR, 2021.

[Szlendak et al., 2021] Permutation compressors for provably faster distributed nonconvex optimization. arXiv preprint arXiv:2110.03300, 2021.

---

### Official Review · Reviewer_xtiL · 2022-10-24

**Confidence:** 2
**Clarity, Quality, Novelty And Reproducibility:** This paper is presented clearly.
**Correctness:** 4
**Technical Novelty And Significance:** 4
**Empirical Novelty And Significance:** 4
**Recommendation:** 6

**Strength And Weaknesses:**

Strength: This paper generalizes the PAGE algorithm and analyzes that it can improve the convergence rate with virtually any (unbiased) sampling mechanism using a novel assumption.It is helpful in the analysis of problems from federated learning. This paper is theoretically sound in general.


**Summary Of The Paper:**

This paper generalizes the PAGE algorithm and analyzes that it can improve the convergence rate with virtually any (unbiased) sampling mechanism using a novel assumption.It is helpful in the analysis of problems from federated learning.Some carefully designed experiments have verified theoretical results.

**Summary Of The Review:**

This paper generalizes the PAGE algorithm and analyzes that it can improve the convergence rate with virtually any (unbiased) sampling mechanism using a novel assumption.It is helpful in the analysis of problems from federated learning.Some carefully designed experiments have verified theoretical results.
Although the paper is theoretically sound, there are still some questions need to be discussed in this paper:
1.	About assumption.It would be better to make a attempt to prove assumption 4 was satisfied in SPIDER and SARAH.
2.	The chart in Figure 3 lacks coordinate description.
3.	About the experiments. All experiments are telling that some samplings can improve the convergence rate of PAGE.Importance sampling performs better.About other sampling schemes in Table 1,it would be better to add them to the experiment.

---

### Official Review · Reviewer_CFoS · 2022-10-26

**Confidence:** 3
**Correctness:** 3
**Technical Novelty And Significance:** 2
**Empirical Novelty And Significance:** 2
**Recommendation:** 5

**Clarity, Quality, Novelty And Reproducibility:**

The writing and presentation are clear. However, the originality and quality are limited.

**Strength And Weaknesses:**

**Strength**:

1. The motivation of doing a refined analysis seems reasonable to me. The authors argue that this could be useful in federated learning.
2. The more detailed analysis does improve the previous results and are more flexible for different sampling methods.
3. The study of different samplings satisfying the weighted AB inequality is interesting, which provides intuitions for differences between sampling methods.

**Weaknesses**:

1. The work is still not enough to convince me doing the analysis does provide enough benefits. For example, from federated learning results in Section A.3 it can be concluded that the importance sampling achieved the best performances. How can we conclude from the complexity results in Table 2 that the analysis does reflect the experimental results?
2. The results in Table 1 seem interesting. As mentioned in the paper, larger B values allow tighter results to be obtained. How do other quantities have impacts on performances of sampling methods?
3. It is unclear to me how should we compare different sampling methods from Table 2. In particular, how do we compare complexity quantities of different samplings.

**Summary Of The Paper:**

This work reconsiders the problem of finding stationary points of finite-sum of $n$ smooth functions, where the optimal complexity for stochastic first-order methods is already known to be $\mathcal{O}(n + n^{1/2} \epsilon^{-1})$. The authors proposed to do a refined analysis since the $\mathcal{O}$ could hide dependencies on smoothness constants, which could be unbalanced across different individual summands, and also a more detailed analysis for different sampling procedure to obtain stochastic gradients.

The authors generalize the analysis for PAGE method (Algorithm 1)to work for different sampling mechanisms, where the $L_+$ smoothness constants could possibly be improved as shown in Table 2. In particular, the authors define a new quantity called weighted Hessian Variance, which improves previous results as also shown in Table 2. Finally, the authors show that the analysis works for samplings used in federated learning.

The key technical results are to show that different sampling methods satisfy a weighted AB inequality (Assumption 4) as summarized in Table 1.

**Summary Of The Review:**

Overall, I found the motivation reasonable and results interesting. However, it is unclear to me how we could use the results to compare different samplings and guide practice, which makes it unclear to me if doing the refined analysis does provide enough benefits.

---

### Official Review · Reviewer_NfNX · 2022-11-05

**Confidence:** 4
**Correctness:** 3
**Technical Novelty And Significance:** 2
**Empirical Novelty And Significance:** 1
**Recommendation:** 3

**Clarity, Quality, Novelty And Reproducibility:**

I feel that the novelty is limited; most results are already known. The only contribution of bringing out the two smoothness constants and showing a sharper rate depending on them. In practice, these parameters are not known and this analysis while a good exercise has limited importance.

**Details Of Ethics Concerns:**





**Strength And Weaknesses:**

#### ***Strengths*** :
1. Overall the paper is really well written, proofs are clean and easy to follow,
2. The refined analysis will serve as a good consolidated reference note for using sampling in practical optimization settings - while the proofs are hard to find from a plethora of different papers.

#### ***Clarifications*** :

1. Overall, I feel that the novelty is marginal. The main novelty in my understanding is the introduction of two smoothness constants introduced in Def 2,3. The rest of the proofs seem routine given the proof technique introduced in Richtárik62and Takáˇc (2016) in the study of randomized coordinate descent methods, Horváth and Richtárik63(2019) and Qian et al. (2021) in analyzing SVRG, SAGA, and SARAH.

2. The paper simply extends the convergence results of PAGE under the non-traditional finer smoothness constants and the results while new are not surprising.

3. The claims rely on the sampling strategy being aware of $L_i$ - I am not sure how is this of any practical importance since it is not available in practice.

4. PAGE with important sampling performs better (Fig 3) - well this is known result.

**Summary Of The Paper:**

#### ***Background*** :
Current convergence results to attain \epsilon approximate stationary point using first-order stochastic methods (ex SGD and its more recent variants like SARAH, PAGE) while achieve the optimal rates matching the lower bounds; the analysis has the following issues - (a) The big-O notation in the above results hides important and typically very large data-dependent constants (smoothness). The paper shows via simple examples to motivate why the dependence on L can be crucial. (b) Further these results are under the 'random' data sampling assumption -- which is often not true in modern distributed and federated learning settings. Prior work has analyzed different non-optimal SGD variants under different sampling schemes and showed improved constants.

#### ***Main Contribution*** :
In this paper the authors unify these proof ideas and propose a unified framework to analyze PAGE and also show that it is possible to improve the convergence under certain sampling assumptions.

**Summary Of The Review:**

I reviewed an earlier version of the paper for NeurIPS and unfortunately, the paper hasn't improved - in fact on a closer look they seem to have submitted the same draft w/o any changes. The paper still has significant weaknesses and unfortunately they didn't use the resubmission to improve upon them.

---

### Decision · Program_Chairs · 2023-01-20

**Decision:**

Reject

**Justification For Why Not Higher Score:**

Lack of novelty especially in technical approaches to the proofs.

**Justification For Why Not Lower Score:**

N/A

**Metareview: Summary, Strengths And Weaknesses:**

As pointed out by various reviewers the novelty of the manuscript is a concern. Specifically, the paper simply extends the convergence results of PAGE under more stringent smoothness constants using techniques which are very similar to the state of the art.
As such I do not feel it included enough novelty for the audience of ICLR.